# Exploring Land System Options to Enhance Fire Resilience under Different Land Morphologies

João Ferreira Silva [1,*], Selma B. Pena [1], Natália S. Cunha [1], Paulo Flores Ribeiro [2], Francisco Moreira [3,4,5] and José Lima Santos [2]

1. Linking Landscape, Environment, Agriculture and Food (LEAF), Associate Laboratory TERRA, School of Agriculture, University of Lisbon, Tapada da Ajuda, 1349-017 Lisbon, Portugal; selmapena@isa.ulisboa.pt (S.B.P.); natcunha@isa.ulisboa.pt (N.S.C.)
2. Forest Research Centre, Associate Laboratory TERRA, School of Agriculture, University of Lisbon, Tapada da Ajuda, 1349-017 Lisbon, Portugal; pfribeiro@isa.ulisboa.pt (P.F.R.); jlsantos@isa.ulisboa.pt (J.L.S.)
3. CIBIO, Centro de Investigação em Biodiversidade e Recursos Genéticos, InBIO Laboratório Associado, Campus de Vairão, Universidade do Porto, 4485-661 Vairão, Portugal; fmoreira@isa.ulisboa.pt
4. CIBIO, Centro de Investigação em Biodiversidade e Recursos Genéticos, InBIO Laboratório Associado, Instituto Superior de Agronomia, Universidade de Lisboa, Tapada da Ajuda, 1349-017 Lisbon, Portugal
5. BIOPOLIS Program in Genomics, Biodiversity and Land Planning, CIBIO, Campus de Vairão, 4485-661 Vairão, Portugal
* Correspondence: joaofsilva@isa.ulisboa.pt

**Abstract:** Fire is the origin of serious environmental and social impacts in Mediterranean-like landscapes, such as those in California, Australia, and southern Europe. Portugal is one of the southern European countries most affected by fire, which has increased in intensity and extent in the recent decades in response to variations in climate, but mostly due to changes in land systems (LSs), characterized by land use and land cover and also by factors such as management intensity, livestock composition, land ownership structure, and demography. Agricultural activities, which contributed to the management of fuel in the overall landscape, were allocated to the most productive areas, while the steepest areas were occupied by extensive areas of shrubland and monospecific forests, creating landscapes of high fire-proneness. These challenging circumstances call for landscape transformation actions focusing on reducing the burned area, but the spatial distribution of LS is highly conditioned by land morphology (LM), which limits the actions (e.g., farming operations) that can be taken. Considering the constraints posed by the LM, this study investigates whether there is a possibility of transforming the landscape by single modifying the LS from more to less fire prone. To better understand landscape–fire relationships, the individual and interactive effects of the LS and LM on burned areas were also analyzed. Even in the more fire-prone LM types, a 40% proportion of agricultural uses in the landscape results in an effective reduction in the burned area.

**Keywords:** fire behavior; fire resilience; land-use planning; Mediterranean

## 1. Introduction

Fire has always been a natural disturbance affecting and shaping Mediterranean-like landscapes, such as those in California, Australia, and southern Europe [1,2]. Amongst the southern European countries, Portugal is one of the most affected by wildfires, with an average burned area of almost 140,000 ha per year between 2010 and 2019 [3].

In recent decades, fire has acquired different characteristics in response to changes in climate, but mostly due to a decline in the landscape mosaic that has historically characterized Portuguese rural areas [4–7]. The abandonment of agricultural activities in mountainous areas, previously linked to active management of fuel in the landscape, was accompanied by a landscape homogenization through shrub encroachment and afforestation with monocultures of non-native species (eucalyptus and maritime pine), causing a

change in fire patterns, from frequent and low intensity to less frequent but more intense and extensive [8]. At the same time, the increase in fire suppression efficiency also encouraged the continuous accumulation of fuel in the landscape, creating even more intense and extensive fires [2]. These changes challenge policymakers and land use planners to develop effective policies to decrease the proportion of burned area in the landscape.

Wildfire susceptibility was recently reassessed [9,10], and the necessity of a landscape redesign to reduce the burned area has already been recognized by the Portuguese Government with the creation of the Landscape Transformation Program [11]. However, information on how and where to transform the landscape to achieve this goal is still scarce.

Land systems (LSs), the classification of different land use and land cover (LULC) based on factors such as vegetation type, land management practices, and socio-economic activities, can have a significant influence on the occurrence, behavior, and impact of fires [12–14]. In Portugal, fire-proneness decreases from shrubland areas to forests, with variations in species, to agricultural areas (annual and permanent crops, pastures, and agroforestry systems) and urban areas [15–18]. Livestock composition can have a dual effect on the burned area [8]. While grazing goats, for example, are often associated with burning for pasture renewal, cattle and sheep tend to be associated with a reduction in the fine fuel load in the landscape [19]. Holdings size constraint management actions [20], associated with the fact that the property is mostly private, can restrict the implementation of landscape-level plans [21]. Higher population densities are linked to higher fire ignitions but also to a higher early detection and effective firefighting; therefore, the extent of fires can be limited [4]. These are important gradients to consider when the aim is to decrease the proportion of burned area with single altering LULC since it is the only variable influencing fire behavior that can be modified at the landscape scale [7,15].

Fire behavior is also directly influenced by land morphology (LM) [22–26] since it promotes radiant energy transfer from low to higher topographical levels [27]. Several studies report that topographic characteristics are the most significant variables affecting burn severity [28,29]. In mainland Portugal, between 1990 and 2017, around 46% of the burned area occurred on hillslopes with a slope greater than 16%, while the flattened areas (valleys bottoms and hilltops) only accounted for about 13%.

In addition to individual effects, other studies [22,23] have explored the relationships between LULC and topography in explaining fire behavior. Despite reaching partial contradictory results, both studies concluded that the proportion of burned area was not independent of slope for any LULC category. Still, how the interaction between LSs and LM affects fire behavior remains poorly understood.

Simultaneously, the spatial distribution of LS is conditioned by LM, leading to the generalization that flatter areas are more suitable for agriculture, while more sloping areas have greater suitability for forests and shrubs [30–36]. Effectively, the difficult access for farming operations on larger slopes, and the consequent increase in the risk of erosion by agriculture, make forest and shrubland the more suitable LULC for these locations [37], but at the cost of greatly increasing the proportion of the burned area.

The main objectives of this study are (i) to investigate differences in fire-proneness across LM and LS types at a national scale; (ii) to analyze the interaction effects between LM and LSs on burned area; and (iii) given the constraints that LM places on the distribution of LSs, identify whether there is scope to transform fire-prone landscapes by modifying LS.

To achieve these objectives, two typologies were created. A LM typology, based on three main landforms: valley bottoms, hillslopes, and hilltops. A LS typology, based on five key dimensions: land use, agricultural management intensity, livestock composition, land ownership structure, and demography. With the identification of homogeneous areas of fire-proneness, this study aims to establish priority areas for landscape transformation actions and the application of common strategies of landscape planning and management.

## 2. Materials and Methods

The study area is mainland Portugal, the most southwestern country on the European continent (Figure 1). With an approximate area of 89,084 km², Portugal is bordered by the Atlantic Ocean to the West and south and by Spain to the north and east.

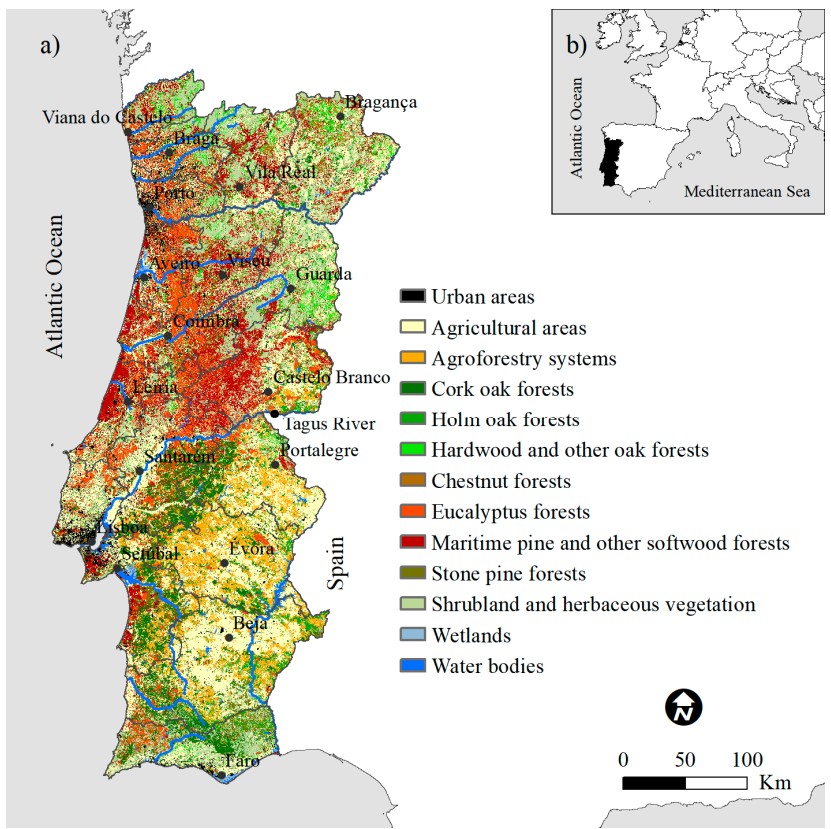

**Figure 1.** Land use and land cover major classes [38] in mainland Portugal (**a**) and an extent with the location of Portugal in Western Europe. (**b**) Administrative districts are identified.

The biophysical environment varies greatly between the areas located north and south of the Tagus River. This variation is largely explained by the marked differences in relief, as most mountainous areas are located in the north (the highest peak is at 1993 m), where total precipitation is close to 2000 mm, and the average annual temperature varies between 6 °C and 15 °C. Except for some mountainous areas in the extreme south, the landscape south of the Tagus River is characterized by a gently waved relief, where the average altitude is around 250 m. In this region, precipitation decreases to approximately 500 mm, and the average annual temperature rises to about 17 °C. During the summer months, the relative humidity is below 70% throughout the mainland territory, dropping below 60% in the hotter inland regions.

Despite this climatic variability between the north and south, Portugal presents typical characteristics of a Mediterranean climate, with the highest temperatures, lowest relative humidity levels, and strongest winds concentrated in the summer period, namely, in July, August, and September, creating the optimal conditions for fire occurrence.

In 2015, agricultural areas (temporary crops, permanent crops, and permanent pastures) accounted for 31% of Portugal [38] (Table 1). If we combine the agroforestry systems, it is obvious that agricultural uses predominate south of the Tagus River. In Portugal, the main agroforestry system is called *montado*, and it is characterized by low tree densities combined with agriculture and/or pastoral activities. The Portuguese *montado* is mainly constituted by two native species: cork oak (*Quercus suber* L.) and holm oak (*Quercus rotundifolia* Lam.), in some areas in combination with stone pine (*Pinus pinea* L.). The maintenance of the *montado* has largely depended on support from the Common Agricultural

Policy (CAP), with a focus on direct support for extensive livestock, with positive effects in terms of its low fire-proneness.

**Table 1.** Distribution of land use and land cover (LULC) classes in mainland Portugal, with emphasis on the differences recorded between the areas north and south of the Tagus River.

| LULC | % of the Area North of the Tagus River | % of the Area South of Tagus River | % in Mainland Portugal |
|---|---|---|---|
| Agricultural areas (temporary crops, permanent crops, and permanent pastures) | 28.05 | 35 | 31.18 |
| Agroforestry systems (cork oak, holm oak, and stone pine) | 0.85 | 17.61 | 8 |
| Chestnut forests (*Castanea sativa* L.) | 1.12 | 0.03 | 0.65 |
| Cork oak (*Quercus suber* L.) | 1.3 | 14.34 | 6.86 |
| Eucalyptus (*Eucalyptus* spp.) | 12.50 | 6.41 | 9.9 |
| Holm oak (*Quercus rotundifolia* Lam.) | 0.85 | 4.17 | 2.26 |
| Maritime pine (*Pinus pinaster* Aiton) | 19.32 | 2.65 | 12.21 |
| Other oaks and hardwood forests (e.g., *Quercus robur* L., *Q. pyrenaica* Willd, and *Q. faginea* Lam.) | 7.47 | 1.03 | 4.72 |
| Stone pine (*Pinus pinea* L.) | 0.27 | 4.97 | 2.27 |
| Shrubland and herbaceous vegetation | 20.01 | 7.51 | 14.68 |
| Urban areas | 6.87 | 2.75 | 5.11 |

Forests and shrubland occupy a large extent of the country (nearly 40% and 15%, respectively) and predominate in the region north of the Tagus River (Table 1). The predominant forest species are non-native maritime pine (*Pinus pinaster* Aiton) (19% of the country) and eucalyptus (Eucalyptus spp.) (13% of the country). The forests of native species such as oaks (e.g., *Quercus robur* L., *Q. pyrenaica Willd*, *Q. faginea* Lam.) are mainly located in less productive areas and/or in steep slopes.

Urban areas account for about 5% of the country, occupying 7% of the area north of the Tagus River and 3% of the south (Figure A2). While the main cities are located along the coast, smaller settlements are dispersed across rural areas, often surrounded by forests and shrublands, and under a high fire-risk.

Most of the land is privately owned and the largest holdings prevail south of the Tagus River, which allowed the scale necessary for the economic viability of agroforestry systems. The northern region, particularly in the interior, is characterized by small holdings, which traditionally supported subsistence agriculture and are currently being abandoned and occupied by shrubs or subject to forest management.

### 2.1. Land Morphology

Land morphology (LM) is a classification of landforms according to their hydrological position in the watershed and typifies two systems in the hillslope profile: wet (concave) and dry (convex) [36]. The wet system consists of streams, water bodies, and valley bottoms, including floodplains, defined as flat or concave areas adjacent to streams with a slope <5%. The dry system encompasses convex slope areas, commonly found on the upper parts of the hillslope profiles. It includes hilltops as convex areas with slopes <5%. The narrower forms correspond to ridgelines and the wider to large hilltops, commonly referred to as plateaus. Hillslopes were classified according to different slopes: 0–12%; 12–16%; 16–25%; and >25%.

The LM map with 25 m spatial resolution, based on flat areas, surface curvature, and hydrological features, was adapted from [36].

### 2.2. Land Systems

We characterized land systems (LSs) from a list of variables at the parish level (the smallest administrative region in Portugal), whose relationship with fire has already been studied [15,22,39–42]. In this study, we considered that Portugal was administratively divided into 4050 parishes, with a variation in the area between 5.15 and 43,527.44 ha

(mean = 2199.61 ha). Currently, the number of parishes is lower, as many smaller parishes have been aggregated.

We compiled the exploratory variables from LULC digital map from 2015 [38], the 2009 Agricultural Census [43], and the 2011 Census of Population and Housing [44]. We further organized these variables into five dimensions (Table 2): "Land use", "Agricultural management intensity", "Livestock composition", "Land ownership structure", and "Demography". We also mapped each variable to understand its spatial distribution in the study area (Figures A1–A3).

**Table 2.** Summary statistics of variables used in the construction of land morphology and land system typologies and corresponding data sources. LSU = livestock units. UAA = utilized agricultural area.

| Code | Description | Min. | Mean | Max. | Data Source |
|---|---|---|---|---|---|
| | 1. Land morphology (proportion of parish area) | | | | |
| VALLEY | Valley bottoms | 0 | 0.111 | 1 | |
| SLOP012 | Slopes between 0 and 12% | 0 | 0.295 | 0.75 | |
| SLOP1216 | Slopes between 12 and 16% | 0 | 0.105 | 0.27 | |
| SLOP1625 | Slopes between 16 and 25% | 0 | 0.161 | 0.54 | [36] |
| SLOP25 | Slopes greater than 25% | 0 | 0.219 | 0.93 | |
| HILLTOP | Hilltops | 0 | 0.108 | 0.92 | |
| | 2. Land systems | | | | |
| | 2.1. Land use | | | | |
| | 2.1.1. Urban areas (proportion of parish area) | | | | |
| URBAN | Urban areas | 0 | 0.131 | 1.03 | |
| | 2.1.2. Farmland (proportion of parish area) | | | | |
| AGRIC | Agricultural areas (temporary crops, permanent crops, permanent pastures) | 0 | 0.309 | 0.96 | |
| AGFOR | Agroforestry systems | 0 | 0.018 | 0.79 | |
| | 2.1.3. Forest and shrubland (proportion of parish area) | | | | [38] |
| CORK | Cork oak forests | 0 | 0.02 | 0.66 | |
| HOLM | Holm oak forests | 0 | 0.006 | 0.43 | |
| OAKHAR | Other oaks and hardwood forests | 0 | 0.072 | 0.64 | |
| CHEST | Chestnut forests | 0 | 0.008 | 0.63 | |
| EUCALYP | Eucalyptus forests | 0 | 0.117 | 0.91 | |
| MARPINE | Maritime pine and other softwood forests | 0 | 0.142 | 0.81 | |
| STNPINE | Stone pine forests | 0 | 0.006 | 0.4 | |
| SHRBHER | Shrubs and herbaceous vegetation | 0 | 0.151 | 0.94 | |
| | 2.2. Agricultural management intensity | | | | |
| PRODUC | Average standard output (EUR) per hectare of total land | 0 | 602.1 | 19,142 | [43] |
| GRAZINT | Average grazing LSU per hectare of total land | 0 | 0.154 | 4.05 | |
| | 2.3. Livestock composition | | | | |
| CATTLE | Share of cattle in total grazing LSU | 0 | 0.48 | 1 | |
| SHEEP | Share of sheep in total grazing LSU | 0 | 0.287 | 1 | [43] |
| GOAT | Share of goats in total grazing LSU | 0 | 0.092 | 1 | |
| EQUINE | Share of equine in total grazing LSU | 0 | 0.101 | 1 | |
| | 2.4. Land ownership structure | | | | |
| AGRHOLD | Average size of agricultural holdings (No. of agricultural holdings per UAA) | 0 | 75.36 | 3794.19 | [43] |
| | 2.5. Demography | | | | |
| POPUL | Population density (No. of inhabitants per km$^2$) | 0.9 | 505.32 | 29,495.4 | [44] |

The "Land use" dimension was organized into three major classes: "Urban areas", "Farmland", and "Forest and shrubland". "Farmland" is composed of agriculture (sum of temporary crops, permanent crops, and permanent pastures) and agroforestry systems. "Forest and shrubland" class is composed of forests, separated by species: cork oak (*Quercus suber* L.), holm oak (*Quercus rotundifolia* Lam.), other oaks and hardwood (e.g., *Quercus robur* L., *Q. pyrenaica* Willd, *Q. faginea* Lam.), chestnut (*Castanea sativa* L.), eucalyptus (*Eucalyptus* spp.), maritime pine (*Pinus pinaster* Aiton) and other softwood, stone pine (*Pinus pinea* L.), and shrubs and herbaceous vegetation. It is important to emphasize that in the elaboration

of the LULC cartography from 2015, burned areas were classified according to the land use that existed before the fire.

The "Agricultural management intensity" is composed of the agricultural holding's productivity (average standard output (EUR) per hectare of total land) and the grazing intensity, i.e., the livestock density. The "Livestock composition" dimension characterizes the presence of livestock by category (e.g., sheep, goats).

The "Land ownership structure" dimension corresponds to the average size of agricultural holdings, calculated by dividing the total number of holdings by the utilized agricultural area (UAA), i.e., by the total area taken up by arable land, permanent grassland, permanent crops, and kitchen gardens used by the holding, regardless of the type of tenure or of whether it is used as a part of common land.

In this study, the "Demography" dimension, whose indicator is "population density", intends to represent the potential for land management. In Portugal, the highest population densities are associated with the largest cities, where there are greater proportions of people of working age, younger, and educated people. On the contrary, less populated parishes have a higher proportion of elderly people and less educated people [45].

*2.3. Fire Data*

The fire database was assembled using the Portuguese Institute for Nature Conservation and Forests (ICNF) data. These data include the total burned area, date of occurrence, and its location for the selected period 1990–2017 in the 4050 parishes of Portugal (currently, this number is lower because some parishes were aggregated). For each parish, we estimated the accumulated burned area (ABA), i.e., the sum of all the areas that burned during the period 1990–2017. This was then expressed as the proportion of the parish area (Figure 2) and used as an indicator of fire proneness. As the size of parishes varies significantly from the north to south of the country, this method reduces possible biases. The highest values of ABA are in the central and northern regions and the extreme south of Portugal.

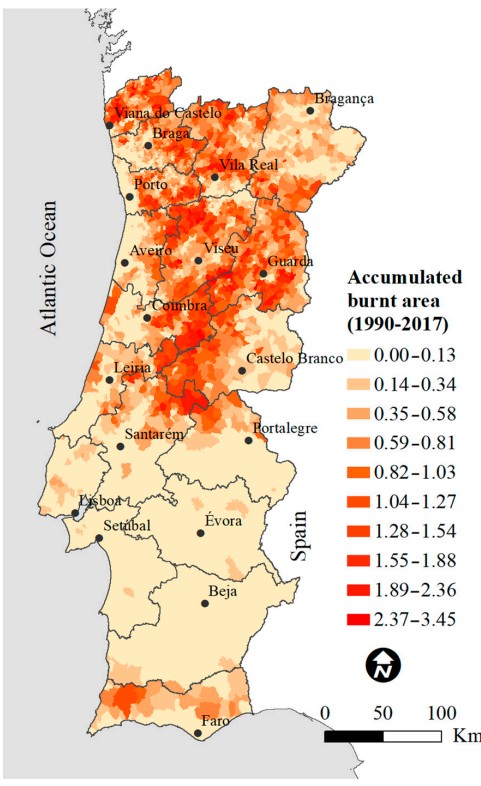

**Figure 2.** Proportion of accumulated burned area for the period 1990–2017 [46], by parish, for mainland Portugal. For example, a value of 3.45 represents parishes that burned about three and a half times their area in this period. Administrative districts are identified.

*2.4. Statistical Analysis*

Land morphology (LM) and land system (LS) typologies were constructed using two unsupervised classification methods using parishes as units of analysis: principal component analysis (PCA) and Ward's hierarchical clustering method (HCA).

The PCA, performed on a correlation matrix, was used to reduce data dimensionality from the LM and LS variables, involving eigenvalue calculation, and selection of principal components based on explained variance. A Varimax orthogonal rotation was applied to minimize the number of variables with high scores. Only factors with eigenvalues greater than 1 were considered in the HCA [47].

The HCA was subsequently applied to cluster the data, with Ward's method serving as the linkage criterion, aiming to minimize the variance within clusters and produce clusters of roughly equal sizes. The clustering results were further evaluated based on the resulting HCA dendrograms and using the "Elbow method", i.e., using a plot of the sum of squared errors (SSE) versus the number of clusters. In clustering contexts, SSE refers to the sum of squared differences between each data point and the centroid of the cluster where the data point belongs. The optimal number of clusters to describe the structure in the dataset was determined by locating a breaking slope ("elbow point") on the plot of the SSE versus the number of clusters. Subsequently, to better understand the results, we mapped the LM and LS types (clusters) using a geographic information system (GIS).

The LM types and LS types were cross-tabulated (contingency table), and a chi-square test was carried out to determine the statistical significance of the relationship between the two classifications.

We used the Shapiro–Wilk test to test data for normality, but this condition was not met. For this reason, we performed the Kruskal–Wallis rank-based nonparametric test and post hoc Dunn's test to determine differences among the fire-proneness (ABA) of both LM types and LS types (significant values $p \leq 0.05$). The adjustments to the $p$-value on Dunn's test were realized with the "Bonferroni" method. We used boxplots to visualize the graphical result.

We also performed a two-way ANOVA to explore whether there was an interaction between the two independent variables (LM types and LS types) on the dependent variable (ABA). Although the analysis of variance (ANOVA) assumes that the data fit the normal distribution, it is not very sensitive to moderate deviations from normality. Several studies, using a variety of non-normal distributions, have shown that the false positive rate is not substantially affected by this violation of the assumption when the samples are large [48–50].

We mapped the various combinations between LM and LS types and their effect on ABA using a GIS to visualize its spatial distribution.

Statistical analyses were carried out with R version 3.5.1 [51], using the following packages: "psych" [52], "nFactors" [53], "FSA" [54], "dunn.test" [55], "rcompanion" [56]. ArcGIS 10.6 software [57] was used for mapping.

## 3. Results

*3.1. Land Morphology Typology*

We obtained two principal rotated components (RCs) with eigenvalues >1.0 (Table A1) from the PCA with Varimax rotation. These two components retained 81% of the variability in the original data.

The resulting HCA dendrogram and the analysis of the plot of the SSE versus number of clusters (Figure A4) suggested a cut-off point of three clusters (Table A3):

- Type I: gently wavy—parishes characterized by the predominance of large valleys interspersed with large hilltops, between which the transition is made with hillslopes with slopes of 0–12%;
- Type II: hilly—parishes characterized by a high proportion of hillslopes with slopes of 0–12%, combined with slopes ranging between 12 and 16%. The valleys and hilltops are narrower than in Type I;

- Type III: steep—parishes characterized by a high proportion of hillslopes with slopes greater than 16%, with a prevalence of slopes above 25%. The valleys and hilltops are narrower than in Types I and II and flat areas are scarce.

The mapping of the three LM types is shown in Figure 3.

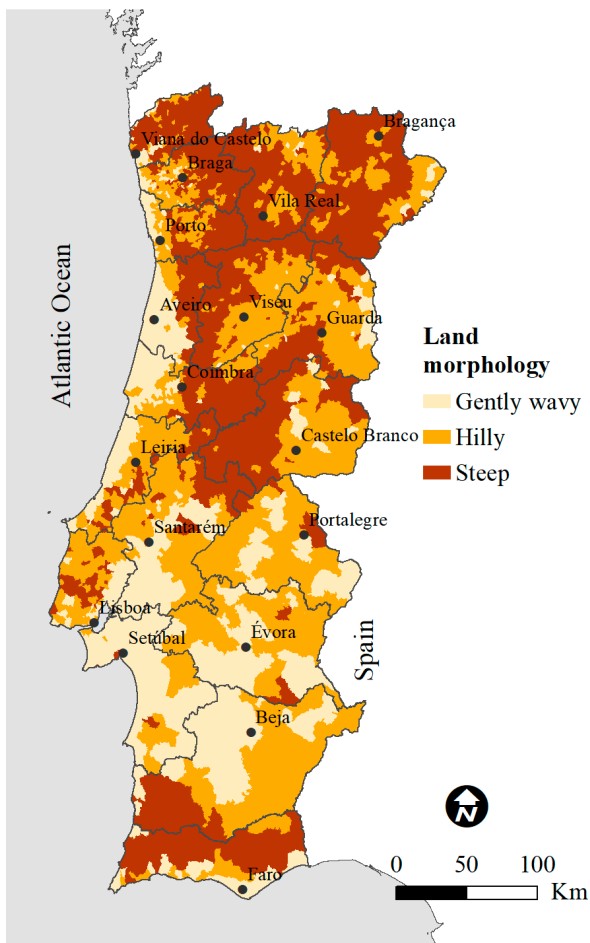

**Figure 3.** Map of the three land morphology types. Administrative districts are identified.

*3.2. Land Systems Typology*

From the principal component analysis with Varimax rotation, performed on the 19 LS variables, we obtained seven principal RCs with eigenvalues >1.0 (Table A2). These seven components retained 66% of the variability in the original data.

The location of the elbow points in the SSE versus the number of clusters (Figure A6) suggested two potential cut-off points: three or eight clusters. As three clusters are not enough to reveal the diversity of LSs that exists in Portugal, we opted for the solution of eight clusters. The eight clusters are characterized based on the summary statistics of background variables (Table A4) and the distribution of these variables by each cluster (Figure A5):

- Type I: maritime pine forests and shrubland grazed by goats (MpiShr)—parishes where land use is dominated by maritime pine forests. Shrubland also occupies a large proportion of the area. Excluding Type VIII, associated with major urban areas, agricultural uses have the lowest proportion in the landscape. It is typified by low agricultural production and grazing intensity. Goats dominate the livestock composition, followed by sheep. The average size of agricultural holdings is small. Population density is the lowest of the eight types.

- Type II: shrubland and other oaks and hardwood forests (ShrOak)—parishes where land use is predominantly composed of shrubland, agriculture, and maritime pine. Concerning Type I, there is a decrease in the proportion of maritime pine at the expense of the increase in agricultural areas and areas of natural regeneration, translated into small oak forests punctuating the extensive areas occupied by shrubs. Despite the small area occupied by forests of other oaks and hardwood and chestnut trees in the country, these species are mostly concentrated in this type. Agricultural production and grazing intensity are low. Cattle dominate the livestock composition, but the proportion of sheep is also relevant. The average size of agricultural holdings is medium, being the second highest, after Type VI. Population density is average for a rural area (289 inhabitants per km$^2$) (Table A4).
- Type III: eucalyptus forests (Eucalyp)—parishes where land use predominantly comprises eucalyptus forests, coexisting with some maritime pine forests. The second most relevant land use is agriculture with a proportion identical to Type II. Agricultural production is low to medium and grazing intensity is medium. It has the second highest proportion of cattle in the eight groups, followed by sheep. The average size of agricultural holdings is one of the smallest. Similar to Type II, population density is average for a rural area.
- Type IV: Mediterranean agriculture (MedAgr)—parishes characterized by a high proportion of agricultural uses (usually permanent crops such as vineyards and olive trees), followed by shrubland. The forest is composed of maritime pine and a significant proportion of native species forests (cork oaks, other oaks and hardwood, and chestnut). Agricultural production is low to medium and grazing intensity is low. The proportion of equine is the largest of the eight types, but the proportion of sheep is also relevant. Excluding Type VIII, associated with major urban areas, the average size of agricultural holdings is the smallest. Population density is average for a rural area.
- Type V: grazing sheep (ShpAgr)—similar to Type IV, agriculture occupies a large proportion of the landscape, sharing it with maritime pine and shrubland. Agricultural production is low to medium, and grazing intensity is low. Sheep dominate the livestock composition. The average size of agricultural holdings is small, and the population density is average for a rural area.
- Type VI: large-scale agriculture (LgScAgr)—parishes where land use is mostly composed of agroforestry systems, with viability in the large size of agricultural holdings. The forests are mainly composed of cork oak, holm oak, and stone pine. Despite agricultural production being one of the lowest, grazing intensity is the second highest of the eight types. Livestock is dominated by cattle, followed by sheep. The population density is the second lowest, after Type I.
- Type VII: intensive agriculture (IntAgr)—parishes with the largest proportions of agriculture and eucalyptus forests. Agricultural production and grazing intensity are the highest of the eight types. The livestock composition is largely dominated by cattle. The average size of agricultural holdings is medium. Population density is the second highest (about 450 inhabitants per km$^2$) after Type VIII, revealing a transition between rural and urban characters.
- Type VIII: urban areas (Urb)—parishes where land use is characterized by the highest proportion of urban areas and, consequently, where population density is also the highest. The urban character of this type translates into a reduced size of holdings and low proportions of agriculture and forests.

Regarding the dominant landscape characteristics, three LS types ("MpiShr"; "ShrOak"; "Eucalyp") have a predominantly forest character ("Total Forest and Shrubland" > 50% and "Total Farmland" < 26%) (Table A4), four LS types ("MedAgr"; "ShpAgr "; "LgScAgr"; "IntAgr") have high proportions of agricultural uses in the landscape ("Total Farmland" > 42%), even when the remaining area is occupied by forest and shrubs ("MedAgr"; "ShpAgr"), and one LS type has a predominantly urban character ("Urb"). The mapping of the eight types is shown in Figure 4.

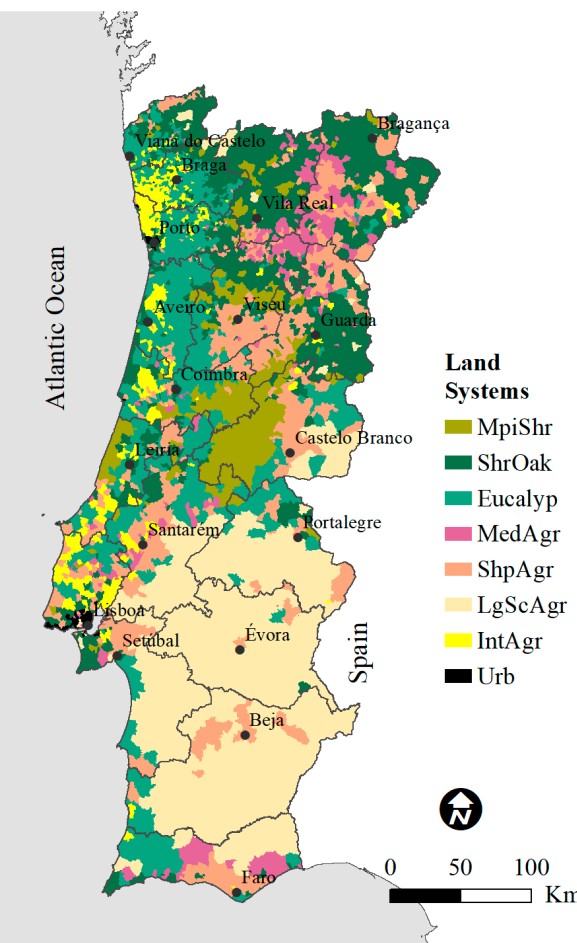

**Figure 4.** Map of the eight land system types. MpiShr: maritime pine forests and shrubland grazed by goats; ShrOak: shrubland and other oaks and hardwood forests; Eucalyp: eucalyptus forests; MedAgr: Mediterranean agriculture; ShpAgr: grazing sheep; LgScAgr: large-scale agriculture; IntAgr: intensive agriculture; Urb: urban areas. Administrative districts are identified.

### 3.3. Association between Land Systems and Land Morphology

The low *p*-value for the $\chi^2$ test ($p < 0.001$) indicates that a statistically significant relationship exists between LM and LS types (Table A5). There is a trend in the distribution of LS types with the highest proportions of "Forest and Shrubland" ("MpiShr", "ShrOak", and "Eucalyp") by the LM types characterized by steeper hillslopes (Figure 5). On the contrary, the LS types with larger "Farmland" ("ShpAgr", "LgScAgr", "IntAgr") and urban areas ("Urb") proportions are associated with less steep LM types. Even so, some of the more agricultural LS types, such as "MedAgr" and "ShpAgr", reveal a strong association with the steeper areas. The highest frequency of the LS type "MedAgr" (60.35%) is found in the LM type Steep.

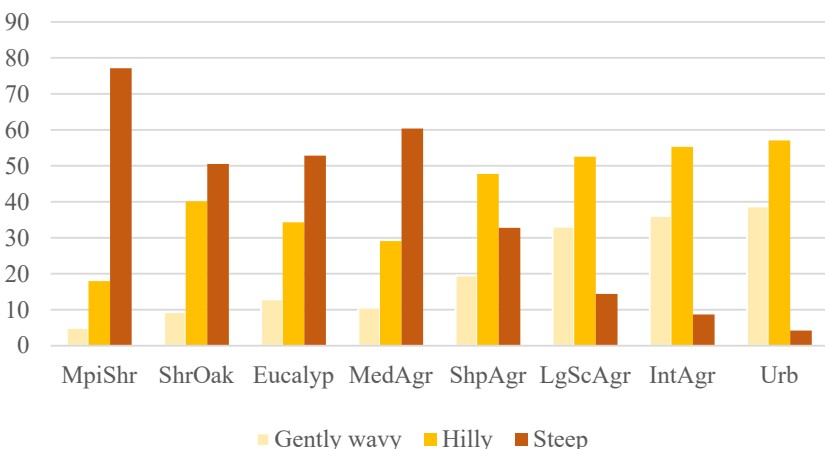

**Figure 5.** Distribution of the eight land system types (MpiShr: maritime pine forests and shrubland grazed by goats; ShrOak: shrubland and other oaks and hardwood forests; Eucalyp: eucalyptus forests; MedAgr: Mediterranean agriculture; ShpAgr: grazing sheep; LgScAgr: large-scale agriculture; IntAgr: intensive agriculture; Urb: urban areas) by the three land morphology types (Gently wavy, Hilly, and Steep).

### 3.4. Association with Fire Regime

3.4.1. Kruskal–Wallis Test

The Kruskal–Wallis rank-based nonparametric test indicated significant differences in LM types concerning ABA ($p$ = <0.001) (Table A6). Overall, fire-proneness increases sequentially with an increase in the proportion of steep slopes in each of the three LM types (Figure 6). The "Gently wavy" LM type is the least fire-prone, followed by LM type "Hilly". The LM type "Steep", characterized by a large proportion of steep hillslopes, is the most fire-prone of all three LM types. All three LM types show significant differences between them.

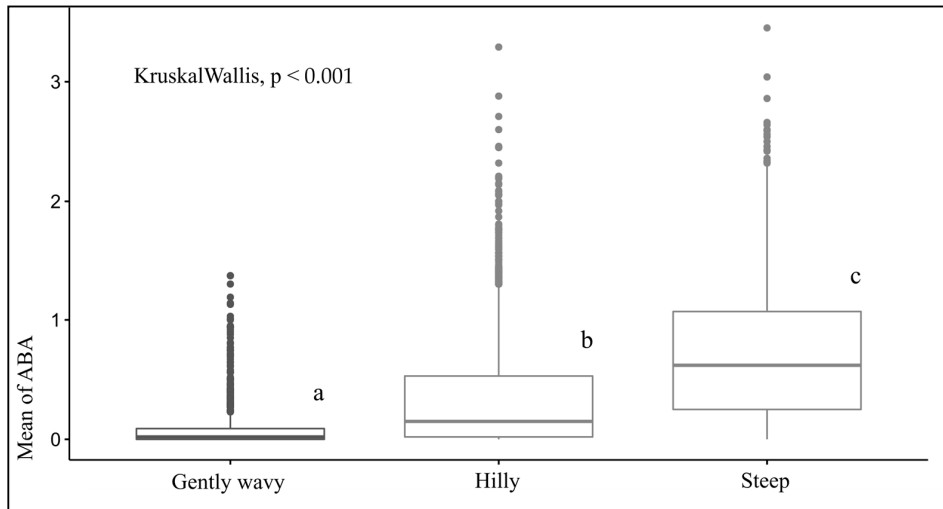

**Figure 6.** Box-and-whisker plots and Kruskal–Wallis' comparisons of the effect of the three land morphology types on the proportion of accumulated burned area (ABA). Within each box, thicker horizontal lines indicate median values; the upper and lower bounds of the boxes indicate the 25th to the 75th percentile of each type's distribution of values; vertical extending lines denote adjacent values; dots refer to observations outside the range of adjacent values; superscript letters report the results of Dunn's pairwise comparisons, where groups with different letters are significantly different. The Kruskal–Wallis test indicated significant differences in LS types concerning accumulated burned area (1990–2017) (ABA) ($p$ = <0.001) (Table A7).

LS types "MpiShr" and "Urb" stand out for being the most and the least fire-prone, respectively, showing significant differences from the other LS types (Figure 7). LS types "ShrOak" and "Eucalyp" are not significantly different from each other and are the second most fire-prone LS type. LS types "MedAgr" and "ShpAgr" are the third most fire-prone and do not show significant differences between them. LS types "LgScAgr" and "IntAgr" are also not significantly different from each other, and they are the second least fire-prone LS types. Overall, excluding LS type "Urb", fire-proneness decreases from LS types with higher proportions of "Total Forest and Shrubland" ("MpiShr"; "ShrOak"; "Eucalyp") to LS types characterized by higher proportions of "Total Farmland" ("MedAgr"; "ShpAgr"; "LgScAgr"; and "IntAgr").

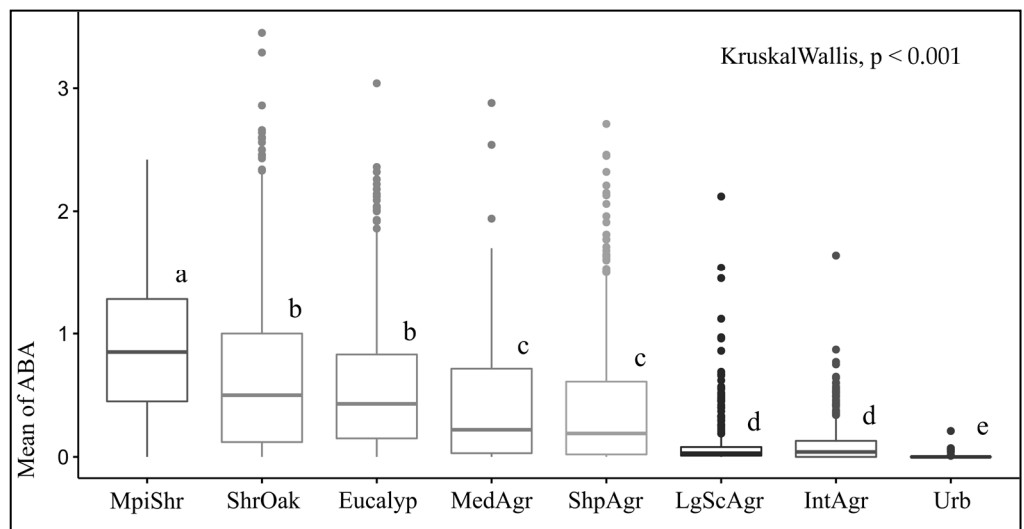

**Figure 7.** Box-and-whisker plots and Kruskal–Wallis comparisons of the effect of land systems on the proportion of accumulated burned area (ABA) for the eight types: MpiShr: maritime pine forests and shrubland grazed by goats; ShrOak: shrubland and other oaks and hardwood forests; Eucalyp: eucalyptus forests; MedAgr: Mediterranean agriculture; ShpAgr: grazing sheep; LgScAgr: large-scale agriculture; IntAgr: intensive agriculture; Urb: urban area. Within each box, thicker horizontal lines indicate median values; the upper and lower bounds of the boxes indicate the 25th to the 75th percentile of each type's distribution of values; vertical extending lines denote adjacent values; dots refer to observations outside the range of adjacent values; superscript letters report the results of Dunn's pairwise comparisons, where groups with different letters are significantly different.

### 3.4.2. Two-Way ANOVA

The two-way ANOVA reveals significant differences between LS and LM types concerning ABA (Table 3). The interaction term (LM:LS) is also statistically significant, which indicates the existence of a significant interaction between LM types and LS types on ABA. The non-parallel lines in the interaction plot (Figure 8) confirm this interactive effect.

**Table 3.** Analysis of variance (two-way ANOVA) of the interaction between land morphology (LM) types and land system (LS) types on the proportion of accumulated burned area. Degrees of freedom, the F-statistic, and *p*-values are shown. Significance code: '****' $p < 0.01$.

|  | Df | Sum Sq | Mean Sq | F Value | Pr (>F) |  |
|---|---|---|---|---|---|---|
| LM | 2 | 216.1 | 108.06 | 522.79 | <0.001 | **** |
| LS | 7 | 90.9 | 12.98 | 62.81 | <0.001 | **** |
| LM:LS | 14 | 22.4 | 1.6 | 7.72 | <0.001 | **** |
| Residuals | 4026 | 832.2 | 0.21 |  |  |  |

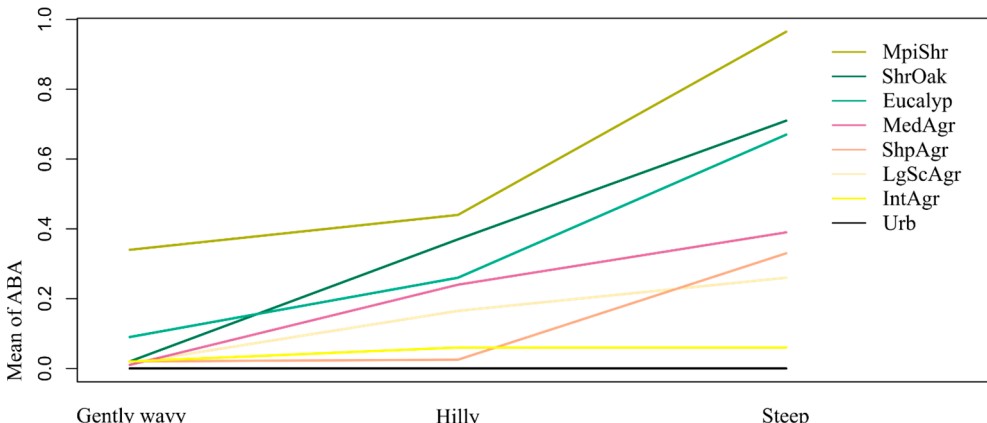

**Figure 8.** Three-way interaction plot for the proportion of accumulated burned area (ABA) as a function of the land morphology types (Gently wavy; Hilly; Steep) and land system types (MpiShr: maritime pine forests and shrubland grazed by goats; ShrOak: shrubland and other oaks and hardwood forests; Eucalyp: eucalyptus forests; MedAgr: Mediterranean agriculture; ShpAgr: grazing sheep; LgScAgr: large-scale agriculture; IntAgr: intensive agriculture; Urb: urban areas).

The different combinations of LM and LS types reveal different effects on ABA. As expected from the individual analyses, there is a clear tendency for ABA to decrease from types with higher proportions of "Total Forest and Shrubland", to types with greater proportions of "Total Farmland" in the landscape and from these to the urban areas (LS type "Urb") and from the LM type "Steep" for the "Gently wavy" type (Table 4).

**Table 4.** Summary statistics (mean) of the proportion of accumulated burned area for the combinations of land morphology (LM) types (rows) and land system (LS) types (columns). The highest values are represented in red, the intermediate values in yellow, and the lowest values in green.

| Land Morphology Types | Land System Types | | | | | | | |
|---|---|---|---|---|---|---|---|---|
| | **MpiShr** | **ShrOak** | **Eucalyp** | **MedAgr** | **ShpAgr** | **LgScAgr** | **IntAgr** | **Urb** |
| Gently wavy | 0.43 | 0.15 | 0.2 | 0.14 | 0.1 | 0.04 | 0.06 | 0.01 |
| Hilly | 0.52 | 0.55 | 0.39 | 0.45 | 0.42 | 0.08 | 0.12 | 0 |
| Steep | 1.01 | 0.82 | 0.75 | 0.46 | 0.53 | 0.41 | 0.18 | 0 |

Despite the tendency observed, there are variations to consider. The LS type "ShrOak" has a higher ABA than "Eucalyp", except for the LM type "Gently wavy". The LS type "ShpAgr" has a higher ABA for the LM type "Steep" than "MedAgr".

Land morphology widens the differentiation between LS types concerning ABA (Figures 8 and A7). Even so, changing the LS type in steeper slopes (most fire-prone LM type) has a greater effect on ABA than the variation shown by LM types over the more fire-prone LS type ("MpiShr"). For example, if in the LM type "Steep" we change the LS type from "IntAgr" to "MpiShr", the values change from an ABA of 0.18 to its maximum value of 1.01 (Table 4). But if we analyze the ABA for the, e.g., LS type "MpiShr" in different LM types, we only reach a variation of 0.58.

The ABA value (Table 4) of LS types characterized by proportions larger than 40% of "Total Farmland" ("MedAgr", "ShpAgr", "LgScAgr", and "IntAgr", excluding "Urb") (Table A4) varies between 0.53 ("ShpAgr") and 0.18 ("IntAgr") in the more fire-prone LM type ("Steep"). The LS types characterized by values lower than 40% of "Total Farmland", where agriculture is replaced by forest and shrubland, are associated with a substantial increase in ABA, which in the case of the LS type "MpiShr" reaches 1.01, i.e., almost twice as much as the more agricultural LS types.

The map (Figure 9) identifies the homogeneous areas regarding LM and LS combinations, ordered by the mean value of ABA. The higher mean values of ABA are concentrated among central and northern regions, associated with the LM-LS combinations "Steep-MpiShr" (1.01), "Steep-ShrOak" (0.82), and "Steep-Eucalyp" (0.75). The least fire-prone areas are mainly associated with LS type "Urb" for every combination with the three LM types. These areas, associated with the main cities, are located along the coast.

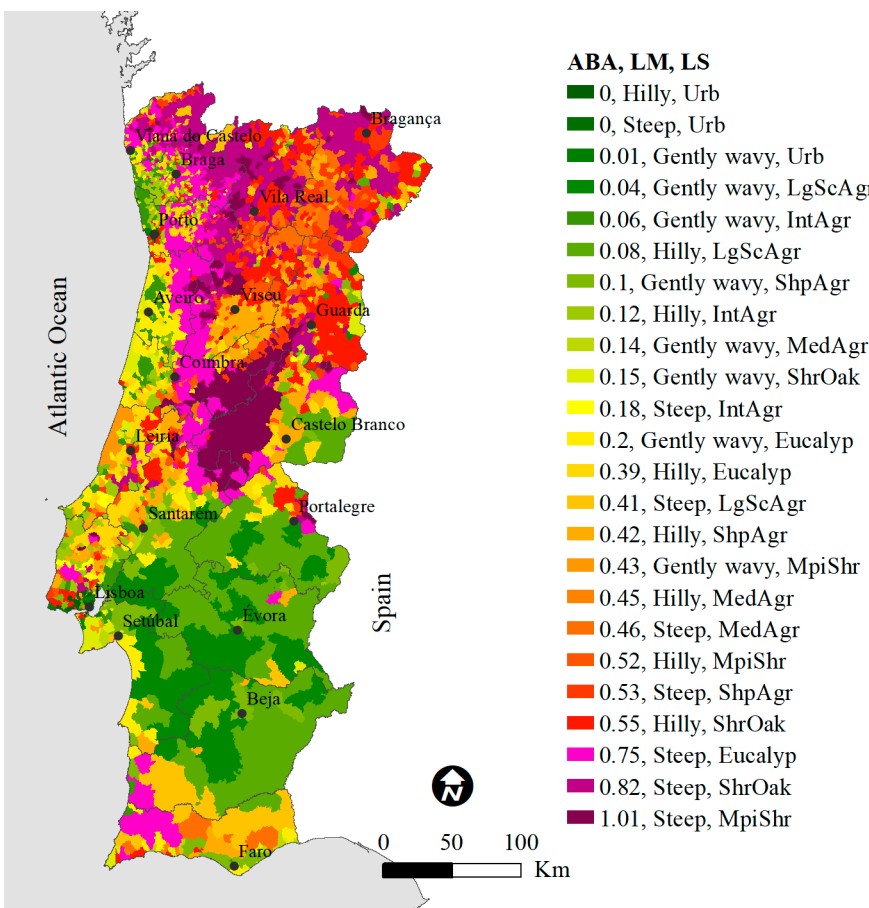

**Figure 9.** Map displaying the spatial distribution of the 24 combinations between land morphology types (Gently wavy; Hilly; Steep) and land system types (MpiShr: maritime pine forests and shrubland grazed by goats; ShrOak: shrubland and other oaks and hardwood forests; Eucalyp: eucalyptus forests; MedAgr: Mediterranean agriculture; ShpAgr: grazing sheep; LgScAgr: large-scale agriculture; IntAgr: intensive agriculture; Urb: urban areas), ordered by the mean value of the proportion of accumulated burned area (ABA). Administrative districts are identified.

The combinations of LS type "LgScAgr" with LM types "Gently wavy" and "Hilly" also have a low fire-proneness and dominate the southern half of Portugal, except for the extreme south, where mean values of ABA increase. Lower fire proneness is also associated with three other agricultural LS types: "IntAgr", "ShpAgr", and "MedAgr", distributed over the LM types "Gently wavy" and "Hilly". The highest mean value of ABA assigned to a LS type with agricultural character is 0.53 and refers to the "Steep-ShpAgr" combination. Yet, this mean value of ABA is about half the LM-LS combination more prone to fire ("Steep-MpiShr").

## 4. Discussion

The climate of Mediterranean regions, characterized by warm–dry summers and cold–humid winters, provides optimal conditions for the rapid growth of vegetation and

the existence of frequent and intense fire events. The mountain landscapes of southern European countries such as Greece and Portugal have been subject to recent phenomena of agricultural abandonment and subsequent shrub encroachment and afforestation with monocultural forests of maritime pine and eucalyptus [58,59]. In addition to changes in land use, other LS dimensions influence the overall fire-proneness of the landscape, such as the average size of holdings, which is smaller in southern European countries than in the other mentioned regions and highly influence the ability to redesign the landscape to reduce the burned area.

The fire-proneness analysis of the various LS types revealed significant differences in line with other studies [15,17,22,23]. The LS types characterized by high proportions of shrubland and forest show greater fire-proneness when compared with LS types characterized by higher proportions of agricultural uses (temporary crops, permanent crops, permanent pastures, and agroforestry systems) and urban areas in the landscape [7]. This gradient may be due to the lower amount of fuel that characterizes agricultural areas and urban areas concerning forests and shrubland. Furthermore, areas with higher proportions of urban areas have higher population densities and, thus, faster detection of ignitions, which, combined with better fighting capacities, enables early fire suppression [19].

Although the land use information [38] used to establish LS types that support fire proneness comparisons corresponds to a temporal snapshot (2015), the fact that the areas that burned just before that moment were classified with the LULC that existed before the fire occurrence (e.g., forest or other, instead of post-fire shrubland) has avoided establishing an erroneous causality between fire historical data (1990–2017) and the LULC (e.g., identifying shrubland as the cause when they would be the consequence of fire).

Fire behavior is also influenced by land morphology (LM). The LM types "Hilly" and "Steep", characterized by high proportions of steep sloping areas, have the highest rate of fire spread [27]. Notwithstanding, LM types induce changes in the LS types' fire-proneness, which generally increases with the increase in the proportion of steep slopes in the landscape.

From the classification of eight LS types, three stand out for their greater fire proneness: "MpiShr", "ShrOak", and "Eucalyp". The LS type "MpiShr" (maritime pine forests and shrubland grazed by goats) reveals the highest fire-proneness, which may be due to the composition of maritime pine (*Pinus pinaster* Aiton) and shrubland, both LULC of high fire-proneness [15,22,60], combined with a predominant distribution in the LM type "Steep" [22,23].

The LS type "ShrOak" (shrubland and other oaks and hardwood forests) also has a high fire-proneness. This type is mostly characterized by areas of natural succession in several stages, where extensive shrubland areas are punctuated by small oak forests, many of them non-mature. Despite the high fire-proneness of this type, the increase in agricultural areas and oak forests and the decrease in maritime pine forests compared with the LS type "MpiShr" negatively affect the burned area. This result is in line with a study that suggests that oak woodlands are generally more fire-resistant than coniferous forests and that an increase in their proportion may result in a decrease in the landscape's susceptibility to fire [6]. Still, under these circumstances, where small oak forests are located in extensive fire-prone areas of shrubland, it is possible that their fire resistance potential is being reduced. Moreover, shrubland is linked to the control of the extension of burned areas in the landscape [60]. Usually, shrublands have the highest number of ignitions (e.g., burning for pastures) [22] and, simultaneously, are considered the least valuable land cover and are given the lowest priority for firefighting. It should be noted that there are variations in the composition of shrubland areas that are not mapped and that will certainly have an impact on fire-proneness. Still, shrublands are essential for biodiversity conservation and the provision of several ecosystem services (e.g., soil protection) [61], and its full elimination is not desirable, but rather its integration into a landscape mosaic. Several studies emphasize the importance of preserving a landscape mosaic to decrease the proportion of burned area in the landscape [1,5,58,62].

The LS type "Eucalyp", characterized by the highest proportion of eucalyptus forests of all LS types, also has high fire-proneness. This may be related to the fact that this LS type is mostly located in the LM type "Steep", with the high flammability of *Eucalyptus* spp. [63], and also because of the large extent occupied by this species in the landscape.

The association between LM types and LS types determines the distribution of LS in the landscape. In general, there is a predominance of LS types with greater proportions of forest (forests and shrubland) in the LM type "Steep" and of LS types with greater proportions of farmland and urban areas in the LM types "Gently wavy" and "Hilly". The constraints placed by LM on LS could suggest that there was little room to transform the landscape to decrease the proportion of burned area, using a single strategy of increasing agricultural proportion. However, our results show that there are LS types of agricultural character of low fire-proneness (e.g., "MedAgr", "ShpAgr") located in LM types with high fire-proneness (e.g., "Steep"). Indeed, LS types linked with livestock grazing [35,64,65] or permanent crops (e.g., vineyards, olive trees), often with terracing practice to prevent soil erosion and facilitate farming operations, are historically adapted to steep areas [66,67].

Despite LM's influence on LS fire-proneness, LS types reveal an even greater effect on the proportion of burned area when we control for the effect of LM. For example, by changing the LS type from "MpiShr" (LS type with highest fire-proneness) to "IntAgr" in the LM type "Steep" (LM type with the highest fire-proneness), the proportion of the burned area is reduced to about one fifth. A landscape transformation at this level, where forests and natural habitats would be replaced with intensive farming systems, could be reflected in a large decrease in the proportion of burned area and an increase of agricultural commodity production, but probably at the expense of a broad range of services, including cultural heritage and identity [68,69], and the loss of biodiversity [70]. In Europe, around 50% of all species rely on agricultural habitats at least to some extent [71], so supporting specific types of low-intensity agriculture would potentially contribute to biodiversity conservation [72]. When located on the wildland–urban interface, the maintenance of low-intensity agricultural areas in steep areas is also essential for the protection of settlements from fire [73].

In general, the proportion of burned area in the landscape decreases as the proportion of agricultural uses increases, suggesting that it is a priority to balance the proportion between forestry (forests and shrubland) and agricultural uses in the landscape, to the detriment of other actions that aim to reduce landscape-scale fire-proneness. Our findings suggest that a proportion of more than 40% of agricultural uses in the landscape (e.g., LS Type "MedAgr", "ShpAgr", "LgScAgr") results in a reduction of about 50% in the burned area in the LM type with the highest fire-proneness ("Steep") (Table 4), compared with landscapes where agriculture only occupies less than 26% (e.g., LS types "MpiShr", "ShrOak", and "Eucalyp").

The LS types are also influenced by the land ownership structure, namely by the size of the holdings [74]. For example, in the LS type "LgScAgr", characterized by the largest average size of holdings, it was possible for farmers to run economically viable agroforestry systems (in Portugal, mainly associated with cork oak and holm oak agroforestry systems). This LS type, which also has a high proportion of cork oak and holm oak forests outside agroforestry systems, reveals a low fire-proneness, particularly when compared with the LS type "ShrOak", where small oak forests are associated with extensive areas of shrubland in small-scale farm areas. Of course, it cannot be ignored that this agro-forest-based LS type predominates in the LM types "Gently wavy" and "Hilly", where there are fewer restrictions on the implementation and management of fuel within the farming systems. It is also worth noting that the LS type "LgScAgr" has the second lowest population density of the eight types and the second lowest fire-proneness, alongside the LS type "IntAgr", revealing that it is possible to create less fire-susceptible landscapes, even in areas with low population density.

To achieve landscapes with lower proportions of burned area, transformation actions should prioritize LM types characterized by large proportions of steep slopes and LS composed of high proportions of shrubland and forests. Here, efforts must be made to increase the proportion of agricultural area, using innovative policy-support solutions and sustainable agricultural practices to make this activity economically viable and, at the same time, avoid negative impacts greater than the fire itself (e.g., soil erosion).

The Landscape Transformation Program [11] identified the territories with the highest fire risk and encouraged the joint action of farmers and forest owners to create a landscape mosaic in these territories, against the payment for ecosystem services. For these actions to be possible, it is necessary to mobilize a substantial part of the Common Agricultural Policy (CAP) funds for these fire-prone areas.

However, in Portugal, as a significant part of CAP support is given depending on the extent of agricultural holdings [75], the large holdings in the South receive most of the funding, with positive results regarding the decrease in the burned area. When distributed based on the extent of the holdings, CAP support fails to consider, e.g., the employment, production or the provisioning of ecosystem services. For the CAP to contribute to reducing the proportion of burned area in the more fire-prone landscapes, it is necessary to partly allocate financial support to the small farms in the less-favored and steep areas, based, e.g., on the provisioning of ecosystem services such as fire protection [76].

Increasing the proportion of agriculture in the landscape can be difficult to achieve at a large scale in the short- to medium-term, or not desirable in some situations (e.g., low suitability areas; nature conservation areas), so parallel actions can be taken to decrease the proportion of burned area in the landscape. As relevant alternatives, we highlight the gradual change in the composition of forests to low-flammability native species, spatially distributed according to the ecological suitability of the land [77] or rewilding initiatives, where unplanned fires under favorable weather conditions, can create new open areas, contributing to biodiversity and increasing fire suppression opportunities at the landscape level [78].

## 5. Conclusions

Fire behavior is influenced by land systems (LSs), and there is a decreasing gradient of fire-proneness from LSs characterized by a higher proportion of shrubland and forests, to those with higher proportions of agriculture and urban areas. Even so, there are differences in fire-proneness regarding the composition of forest and agricultural spaces. In turn, the LS distribution is influenced by land morphology (LM) which, in parallel, also influences fire behavior, with landscapes composed of higher proportions of steep slopes revealing greater fire-proneness.

Although LM constrains the spatial distribution of LS, with more fire-prone LS dominating in steeper regions, there are less fire-prone LS adapted to these morphological conditions (e.g., Mediterranean agriculture, grazing sheep).

The use of typologies allowed the identification of homogeneous areas of LM and LS for the application of common strategies of landscape planning and management. Actions to transform the landscape must prioritize regions characterized by high proportions of steep slopes, whose dominant composition is forest and shrubland. A balance between forest and agricultural uses must be encouraged to achieve a fire-resilient landscape. The results of this study suggest that a proportion of about 40% of agricultural uses (temporary crops, permanent crops, permanent pastures, and agroforestry systems) in the landscape results in a reduction of about 50% in the burned area, compared with landscapes where agriculture only occupies less than 25%.

The European Union's Common Agricultural Policy (CAP) is a financing instrument that addresses natural hazards, including fire, and its funds can contribute to promoting the development of agriculture (e.g., Mediterranean agriculture, grazing) in areas with greater fire-proneness, also focusing on the wildland–urban interface, recovering the traditional agricultural belt around the villages.

**Author Contributions:** Conceptualization, J.F.S., J.L.S., S.B.P. and N.S.C.; methodology, J.F.S., J.L.S. and F.M.; investigation, J.F.S.; formal analysis, J.F.S.; writing—original draft preparation, J.F.S.; writing—review and editing, J.L.S., S.B.P., N.S.C., P.F.R. and F.M.; supervision, J.L.S.; Project administration, S.B.P. All authors have read and agreed to the published version of the manuscript.

**Funding:** This research was funded by National Funds through FCT—Fundação para a Ciência e a Tecnologia, I.P., under the projects UIDB/04129/2020 of the Research Unit LEAF—Linking Landscape, Environment, Agriculture and Food, UIDB/00239/2020 of the Forest Research Centre and João Ferreira Silva (SFRH/BD/109814/2015).

**Institutional Review Board Statement:** Not applicable.

**Data Availability Statement:** The data presented in this study are available on request from the corresponding author.

**Acknowledgments:** This paper is a result of the project "SCAPEFIRE—A sustainable landSCAPE planning model for rural FIREs prevention" (PCIF/MOS/0046/2017), also supported by FCT. J.L.S., F.M., and P.F.R. are thankful for the support received from the FCT through the project "People and Fire" (PCIF/AGT/0136/2017).

**Conflicts of Interest:** The authors declare no conflict of interest.

## Appendix A

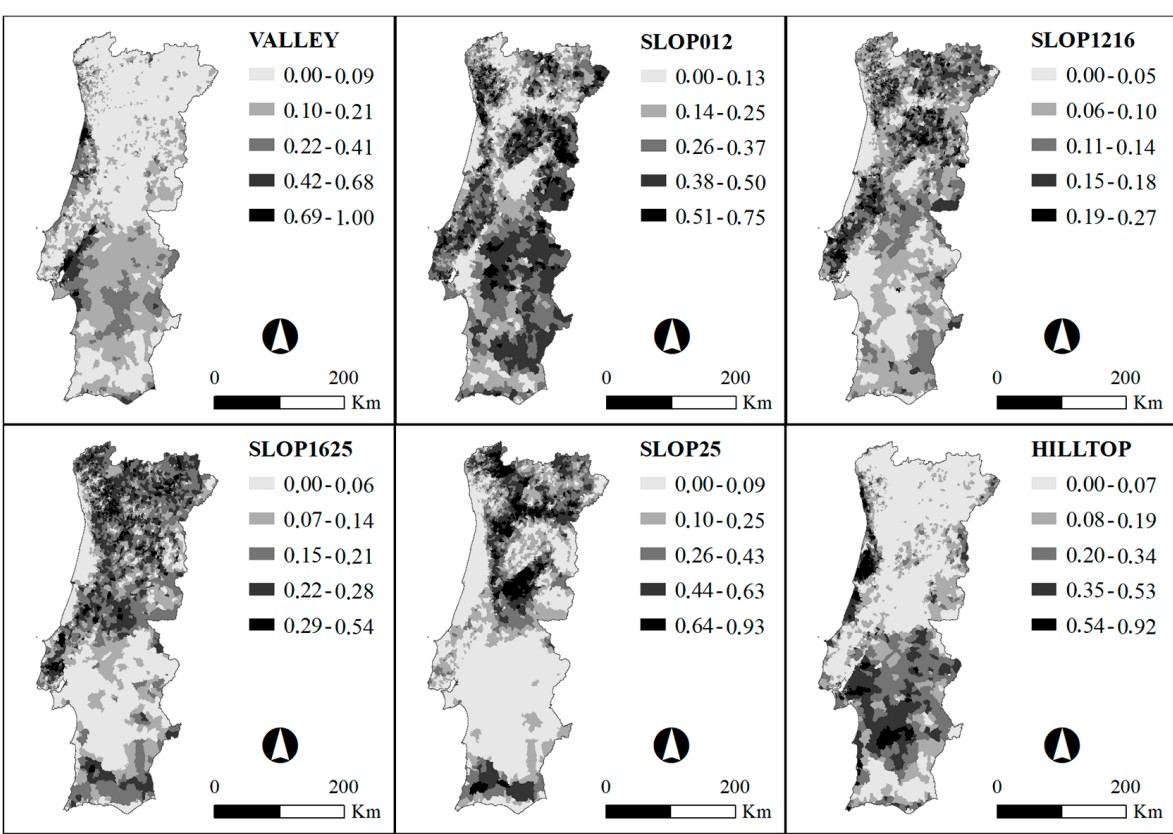

**Figure A1.** Maps of land morphology variables. VALLEY: valley bottoms; SLOP012: slopes 0–12%; SLOP1216: slopes 12–16%; SLOP1625: slopes 16–25%; SLOP25: slopes > 25%; HILLTOP: hilltops.

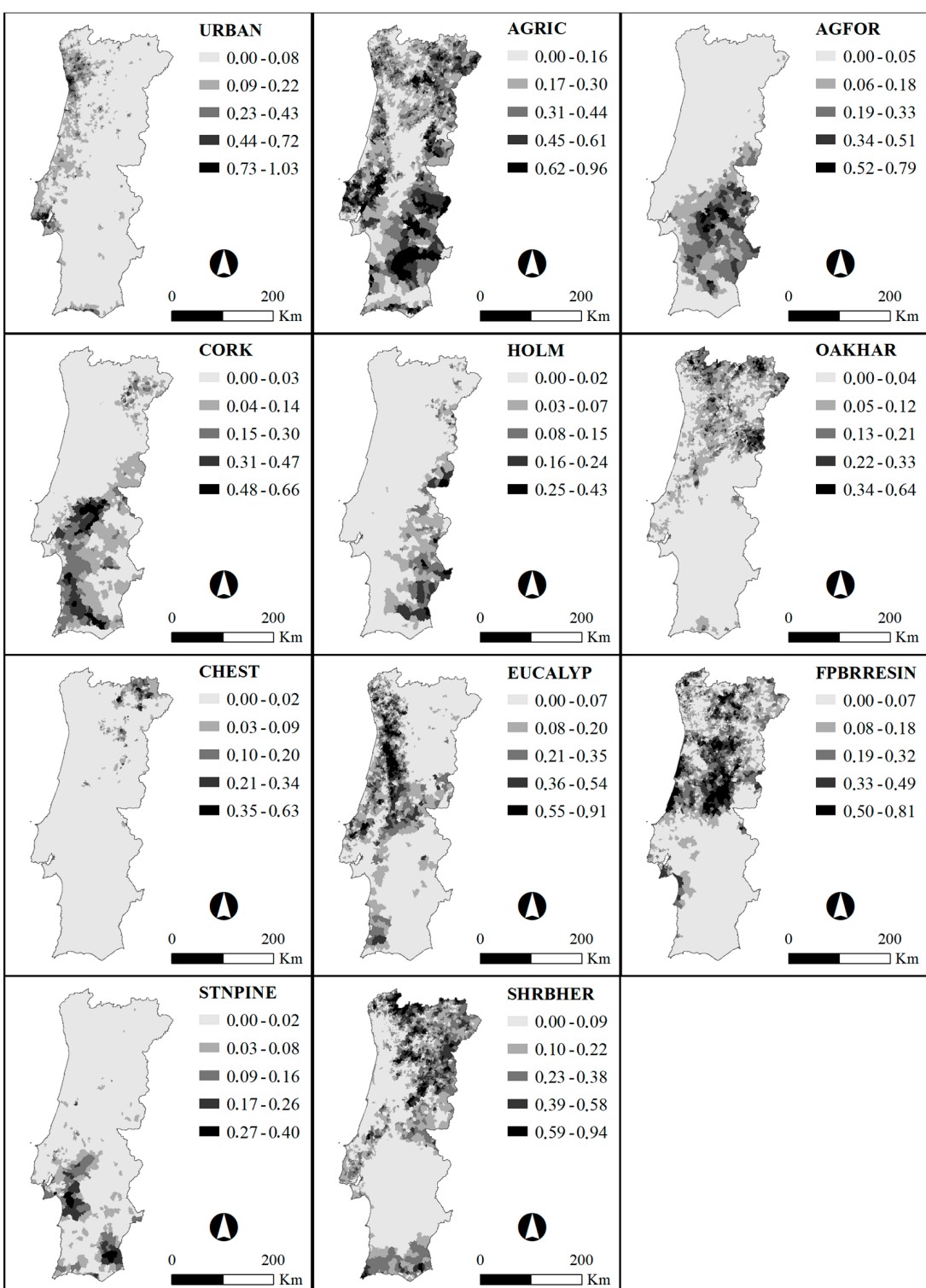

**Figure A2.** Maps of land system variables. URBAN: urban areas; AGRIC: agricultural areas; AGFOR: agroforestry systems; CORK: cork oak forests; HOLM: holm oak forests; OAKHAR: other oaks and hardwood forests; CHEST: chestnut forests; EUCALYP: eucalyptus forests; MARPINE: maritime pine and other softwood forests; STNPINE: stone pine forests; SHRBHER: shrubs and herbaceous vegetation.

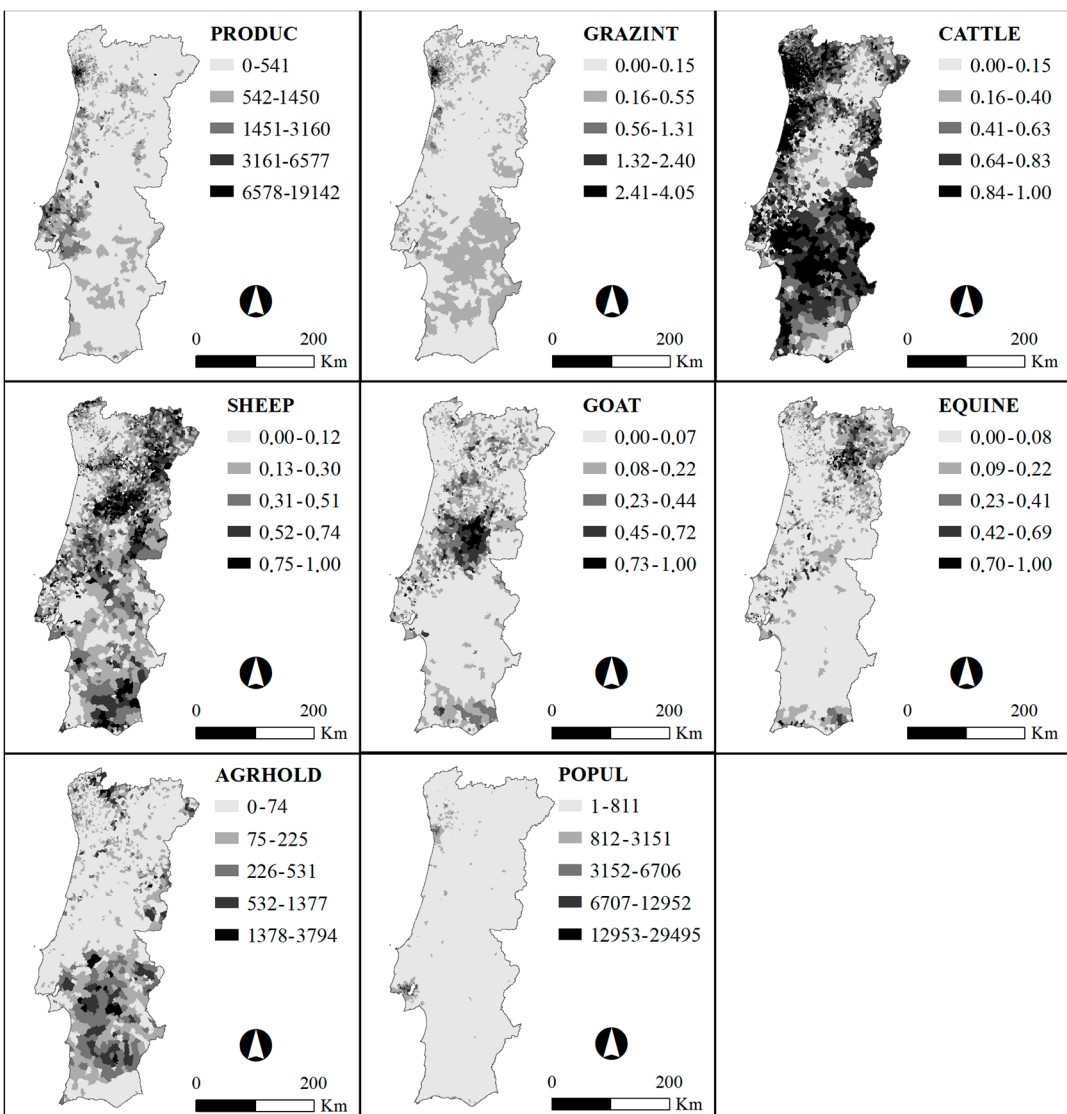

**Figure A3.** Maps of land system variables. PRODUC: average standard output (EUR) per hectare of total land; GRAZINT: average grazing LSU per hectare of total land; CATTLE: share of cattle in total grazing LSU; SHEEP: share of sheep in total grazing LSU; GOAT: share of goats in total grazing LSU; EQUINE: share of equine in total grazing LSU; AGRHOLD: average size of agricultural holdings (No. of agricultural holdings per UAA); POPUL: population density (No. of inhabitants per km$^2$).

**Table A1.** Results of the principal components analysis for each land morphology (LM) type. Principal component analysis performed with Varimax rotation.

| LM | Description | RC1 | RC2 |
|---|---|---|---|
| VALLEY | Valley bottoms | −0.74 | 0.05 |
| SLOP012 | Hillslopes with slopes between 0 and 12% | 0.08 | 0.93 |
| SLOP1216 | Hillslopes with slopes between 12 and 16% | 0.84 | 0.46 |
| SLOP1625 | Hillslopes with slopes between 16 and 25% | 0.90 | −0.21 |
| SLOP25 | Hillslopes with slopes greater than 25% | 0.36 | −0.89 |
| HILLTOP | Hilltops | −0.81 | 0.26 |
| | SS Loadings | 2.85 | 2 |
| | Proportion Variance | 0.48 | 0.33 |
| | Cumulative Variance | 0.48 | 0.81 |

**Table A2.** Results of the principal components analysis for land system (LS) classificatory variables. Principal component analysis performed with Varimax rotation.

| LS | Description | RC3 | RC2 | RC1 | RC5 | RC4 | RC6 | RC7 |
|---|---|---|---|---|---|---|---|---|
| URBAN | Share of urban areas in total land area | −0.17 | −0.9 | −0.02 | 0.1 | −0.19 | −0.16 | −0.08 |
| AGRIC | Share of agricultural areas (temporary crops, permanent crops, permanent pastures) in total land area | 0.06 | 0.28 | 0.65 | −0.27 | −0.05 | −0.4 | 0.14 |
| AGFOR | Share of agroforestry systems in total land area | 0.78 | 0.08 | 0.04 | 0 | −0.06 | −0.16 | −0.16 |
| CORK | Share of cork oak forests in total land area | 0.51 | 0.15 | −0.16 | 0.02 | −0.23 | −0.11 | 0.35 |
| HOLM | Share of holm oak forests in total land area | 0.66 | 0.04 | 0 | −0.14 | −0.02 | −0.09 | −0.1 |
| OAKHAR | Share of other oaks and hardwood forests in total land area | −0.24 | 0.22 | −0.2 | 0.06 | 0.65 | −0.12 | −0.16 |
| CHEST | Share of chestnut forests in total land area | −0.12 | 0.11 | −0.08 | −0.14 | 0.38 | −0.18 | 0.01 |
| EUCALYP | Share of eucalyptus forests in total land area | −0.29 | 0.28 | −0.25 | 0.25 | −0.68 | −0.05 | −0.25 |
| MARPINE | Share of maritime pine and other softwood forests in total land area | −0.21 | 0.16 | 0 | −0.11 | 0.04 | 0.71 | −0.07 |
| STNPINE | Share of stone pine forests in total land area | 0.48 | −0.08 | −0.09 | 0.01 | −0.19 | 0.03 | 0.29 |
| SHRBHER | Share of shrubs and herbaceous vegetation in total land area | 0.02 | 0.11 | −0.23 | 0.04 | 0.65 | 0.22 | 0.21 |
| PRODUC | Average standard output (EUR) per hectare of total land | −0.1 | 0.03 | 0.81 | 0.14 | −0.17 | −0.02 | −0.01 |
| GRAZINT | Average grazing LSU per hectare of total land | 0.01 | −0.01 | 0.75 | 0.34 | −0.06 | −0.02 | −0.16 |
| CATTLE | Share of cattle in total grazing LSU | 0.06 | 0.16 | 0.23 | 0.75 | −0.1 | −0.33 | −0.32 |
| SHEEP | Share of sheep in total grazing LSU | 0.04 | 0.18 | −0.13 | −0.86 | 0.09 | 0.04 | −0.1 |
| GOAT | Share of goats in total grazing LSU | −0.03 | 0.09 | −0.14 | −0.1 | −0.09 | 0.77 | 0.03 |
| EQUINE | Share of equine in total grazing LSU | −0.15 | 0.1 | −0.02 | −0.07 | 0.19 | −0.05 | 0.83 |
| AGRHOLD | Average size of agriculture holdings | 0.59 | 0.09 | 0.07 | 0.3 | 0.25 | 0.03 | −0.11 |
| POPUL | Population density per km$^2$ | −0.04 | −0.92 | −0.11 | 0 | −0.05 | −0.09 | −0.03 |
| | SS Loadings | 2.16 | 2.03 | 1.94 | 1.76 | 1.75 | 1.56 | 1.24 |
| | Proportion Variance | 0.11 | 0.11 | 0.1 | 0.09 | 0.09 | 0.08 | 0.07 |
| | Cumulative Variance | 0.11 | 0.22 | 0.32 | 0.42 | 0.51 | 0.59 | 0.66 |

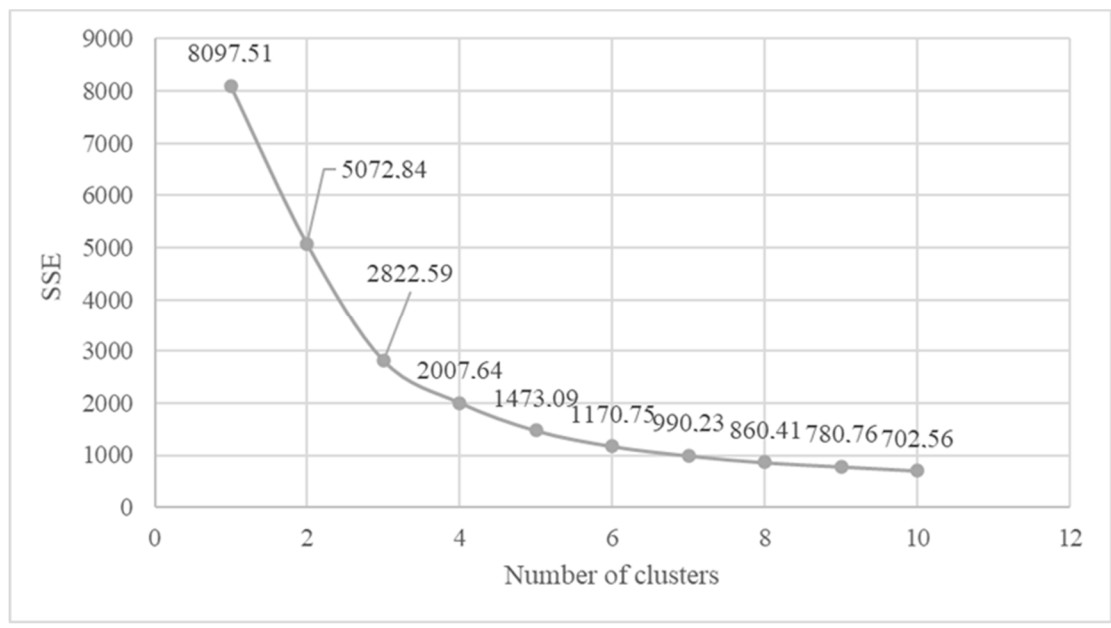

**Figure A4.** Sum of squared error (SSE) versus number of clusters regarding land morphology variables.

**Table A3.** Summary statistics (mean) of background variables in relation to the land morphology (LM) types. The most relevant values are highlighted in bold. *n* = number of observations.

| LM Variables | Description | Gently Wavy *n* = 703 | Hilly *n* = 1675 | Steep *n* = 1672 |
|---|---|---|---|---|
| VALLEY | Valley bottoms | 0.3 | 0.09 | 0.05 |
| SLOP012 | Hillslopes with slopes between 0 and 12% | 0.24 | 0.44 | 0.17 |
| SLOP1216 | Hillslopes with slopes between 12 and 16% | 0.04 | 0.14 | 0.1 |
| SLOP1625 | Hillslopes with slopes between 16 and 25% | 0.04 | 0.15 | 0.22 |
| SLOP25 | Hillslopes with slopes greater than 25% | 0.03 | 0.08 | 0.44 |
| HILLTOP | Hilltops | 0.34 | 0.1 | 0.02 |

**Table A4.** Summary statistics (mean) of background variables in relation to the land system (LS) types: MpiShr: maritime pine forests and shrubland grazed by goats; ShrOak: shrubland and other oaks and hardwood forests; Eucalyp: eucalyptus forests; MedAgr: Mediterranean agriculture; ShpAgr: grazing sheep; LgScAgr: large-scale agriculture; IntAgr: intensive agriculture; Urb: urban areas. "Total Forest and Shrubland" corresponds to the sum of all forest species (cork, holm, oaks and hardwood, chestnut, eucalyptus, maritime pine, and stone pine) and shrubland; "Total Farmland" corresponds to the sum of agriculture (temporary crops, permanent crops, and permanent pastures) and agroforestry systems. *n* = number of observations.

| LS Variables | MpiShr *n* = 223 | ShrOak *n* = 1049 | Eucalyp *n* = 956 | MedAgr *n* = 227 | ShpAgr *n* = 748 | LgScAgr *n* = 278 | IntAgr *n* = 427 | Urb *n* = 142 |
|---|---|---|---|---|---|---|---|---|
| Land Use | | | | | | | | |
| Urban areas | | | | | | | | |
| URBAN | 0.04 | 0.1 | 0.13 | 0.06 | 0.08 | 0.02 | 0.21 | 0.85 |
| Farmland | | | | | | | | |
| AGRIC | 0.13 | 0.26 | 0.25 | 0.42 | 0.44 | 0.35 | 0.45 | 0.03 |
| AGFOR | 0 | 0 | 0 | 0 | 0.01 | 0.21 | 0 | 0 |
| Total Farmland | 0.13 | 0.26 | 0.25 | 0.42 | 0.45 | 0.56 | 0.45 | 0.03 |
| Forest and shrubland | | | | | | | | |
| CORK | 0 | 0 | 0.01 | 0.05 | 0.02 | 0.14 | 0 | 0 |
| HOLM | 0 | 0 | 0 | 0 | 0 | 0.06 | 0 | 0 |
| OAKHAR | 0.06 | 0.14 | 0.05 | 0.07 | 0.06 | 0.01 | 0.04 | 0.02 |
| CHEST | 0.01 | 0.02 | 0 | 0.02 | 0.01 | 0 | 0 | 0 |
| EUCALYP | 0.09 | 0.03 | 0.33 | 0.03 | 0.04 | 0.04 | 0.13 | 0 |
| MARPINE | 0.44 | 0.15 | 0.12 | 0.12 | 0.16 | 0.01 | 0.1 | 0.01 |
| STNPINE | 0 | 0 | 0 | 0.01 | 0.01 | 0.05 | 0 | 0 |
| SHRBHER | 0.23 | 0.27 | 0.08 | 0.2 | 0.15 | 0.07 | 0.05 | 0.06 |
| Total Forest and Shrubland | 0.83 | 0.61 | 0.59 | 0.5 | 0.45 | 0.38 | 0.32 | 0.09 |
| Agricultural management intensity | | | | | | | | |
| PRODUC | 163.84 | 307.46 | 430.47 | 535.84 | 602.77 | 344.17 | 2336.92 | 12.73 |
| GRAZINT | 0.03 | 0.1 | 0.11 | 0.03 | 0.07 | 0.17 | 0.71 | 0.01 |
| Livestock composition | | | | | | | | |
| CATTLE | 0.2 | 0.52 | 0.65 | 0.09 | 0.18 | 0.65 | 0.9 | 0.07 |
| SHEEP | 0.24 | 0.26 | 0.21 | 0.27 | 0.63 | 0.27 | 0.05 | 0.09 |
| GOAT | 0.5 | 0.07 | 0.08 | 0.06 | 0.08 | 0.04 | 0.03 | 0.04 |
| EQUINE | 0.07 | 0.12 | 0.05 | 0.59 | 0.08 | 0.03 | 0.02 | 0.01 |
| Land ownership structure | | | | | | | | |
| AGRHOLD | 58.65 | 80.51 | 39.87 | 32.52 | 40.3 | 372.61 | 65.41 | 3.47 |
| Demography | | | | | | | | |
| POPUL | 50.04 | 288.9 | 240.31 | 170.18 | 192.67 | 92.05 | 454.62 | 7747.29 |

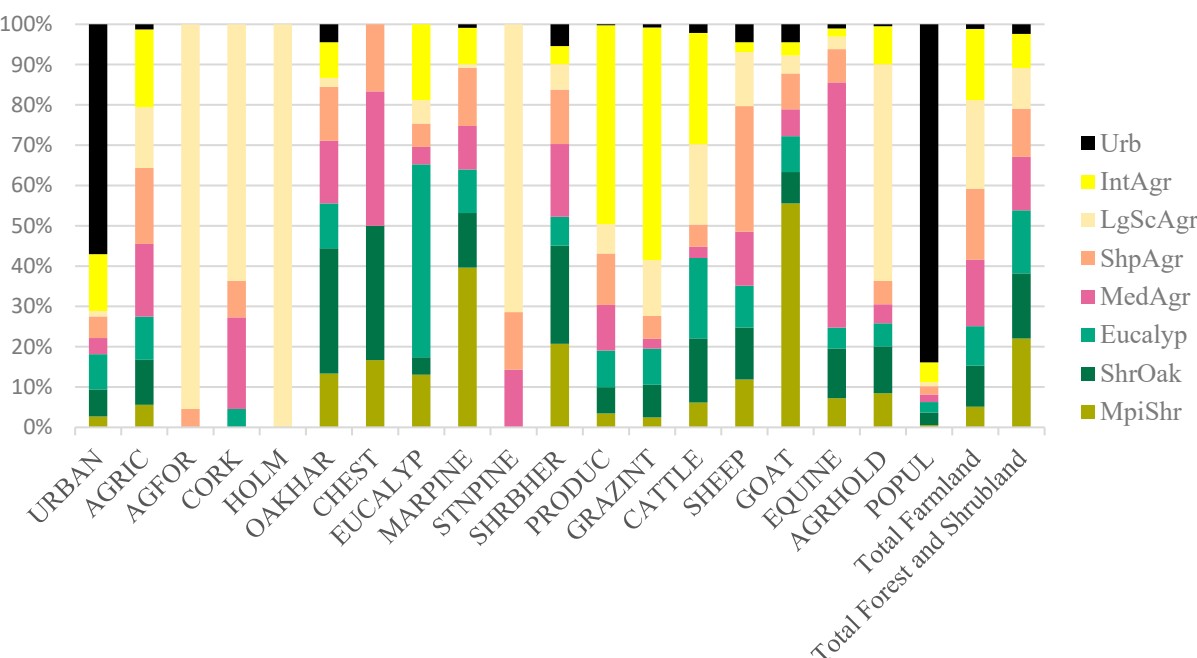

**Figure A5.** Background variable distribution by each of the eight land system (LS) types: MpiShr: maritime pine forests and shrubland grazed by goats; ShrOak: shrubland and other oaks and hardwood forests; Eucalyp: eucalyptus forests; MedAgr: Mediterranean agriculture; ShpAgr: grazing sheep; LgScAgr: large-scale agriculture; IntAgr: intensive agriculture; Urb: urban areas. "Total Forest and Shrubland" corresponds to the sum of all forest species (cork, holm, oaks and hardwood, chestnut, eucalyptus, maritime pine, and stone pine) and shrubland; "Total Farmland" corresponds to the sum of agriculture (temporary crops, permanent crops, and permanent pastures) and agroforestry systems.

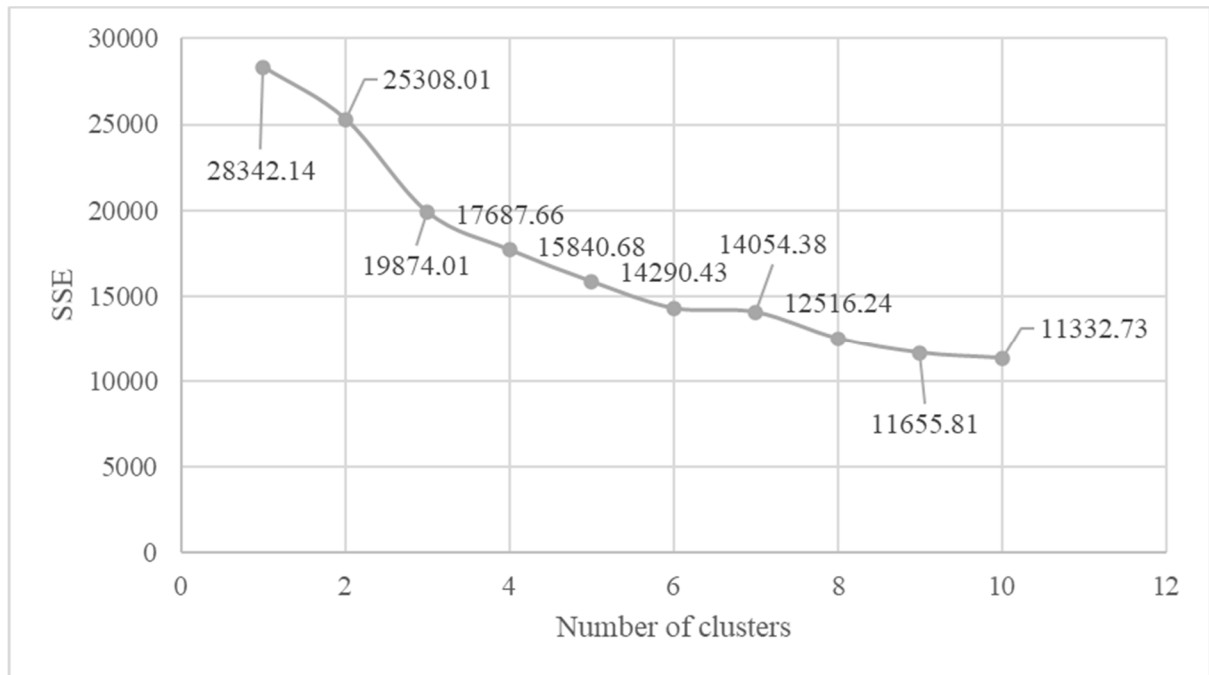

**Figure A6.** Sum of squared error (SSE) versus number of clusters regarding land system classificatory variables.

**Table A5.** Contingency (cross-tabulation) table containing land system (LS) types (columns) by land morphology types (LM) (rows). Each cell of the contingency table contains the number of parishes where a socioeconomic group intersects a biophysical group. Results of the chi-squared test of independence used to determine if there is a significant relationship between land morphology (LM) and land system (LS) types are presented in the final row. MpiShr: maritime pine forests and shrubland grazed by goats; ShrOak: shrubland and other oaks and hardwood forests; Eucalyp: eucalyptus forests; MedAgr: Mediterranean agriculture; ShpAgr: grazing sheep; LgScAgr: large-scale agriculture; IntAgr: intensive agriculture; Urb: urban areas.

| Types | MpiShr | ShrOak | Eucalyp | MedAgr | ShpAgr | LgScAgr | IntAgr | Urb | Sum |
|---|---|---|---|---|---|---|---|---|---|
| Gently wavy | 11 | 98 | 123 | 24 | 146 | 92 | 154 | 55 | 703 |
| Hilly | 40 | 421 | 328 | 66 | 357 | 146 | 236 | 81 | 1675 |
| Steep | 172 | 530 | 505 | 137 | 245 | 40 | 37 | 6 | 1672 |
| Sum | 223 | 1049 | 956 | 227 | 748 | 278 | 427 | 142 | 4050 |
| $\chi^2 = 695.9$, *p*-value < 0.001 | | | | | | | | | |

**Table A6.** Multiple pairwise comparison between land morphology types using the Wilcoxon method. Significance code: '****' $p < 0.01$.

| Type 1 | Type 2 | *p*. Format | *p*. Signif |
|---|---|---|---|
| Gently wavy | Hilly | <0.001 | **** |
| Gently wavy | Steep | <0.001 | **** |
| Hilly | Steep | <0.001 | **** |

**Table A7.** Multiple pairwise-comparisons among land systems using Wilcoxon method. Significance codes: '****' $p < 0.01$; 'ns' = non-significant.

| Type 1 | Type 2 | *p*. Format | *p*. Signif |
|---|---|---|---|
| MpiShr | ShrOak | <0.001 | **** |
| MpiShr | Eucalyp | <0.001 | **** |
| MpiShr | MedAgr | <0.001 | **** |
| MpiShr | ShpAgr | <0.001 | **** |
| MpiShr | LgScAgr | <0.001 | **** |
| MpiShr | IntAgr | <0.001 | **** |
| MpiShr | Urb | <0.001 | **** |
| ShrOak | Eucalyp | 0.13 | ns |
| ShrOak | MedAgr | <0.001 | **** |
| ShrOak | ShpAgr | <0.001 | **** |
| ShrOak | LgScAgr | <0.001 | **** |
| ShrOak | IntAgr | <0.001 | **** |
| ShrOak | Urb | <0.001 | **** |
| Eucalyp | MedAgr | <0.001 | **** |
| Eucalyp | ShpAgr | <0.001 | **** |
| Eucalyp | LgScAgr | <0.001 | **** |
| Eucalyp | IntAgr | <0.001 | **** |
| Eucalyp | Urb | <0.001 | **** |
| MedAgr | ShpAgr | 0.21 | ns |
| MedAgr | LgScAgr | <0.001 | **** |
| MedAgr | IntAgr | <0.001 | **** |
| MedAgr | Urb | <0.001 | **** |
| ShpAgr | LgScAgr | <0.001 | **** |
| ShpAgr | IntAgr | <0.001 | **** |
| ShpAgr | Urb | <0.001 | **** |
| LgScAgr | IntAgr | 0.46 | ns |
| LgScAgr | Urb | <0.001 | **** |
| IntAgr | Urb | <0.001 | **** |

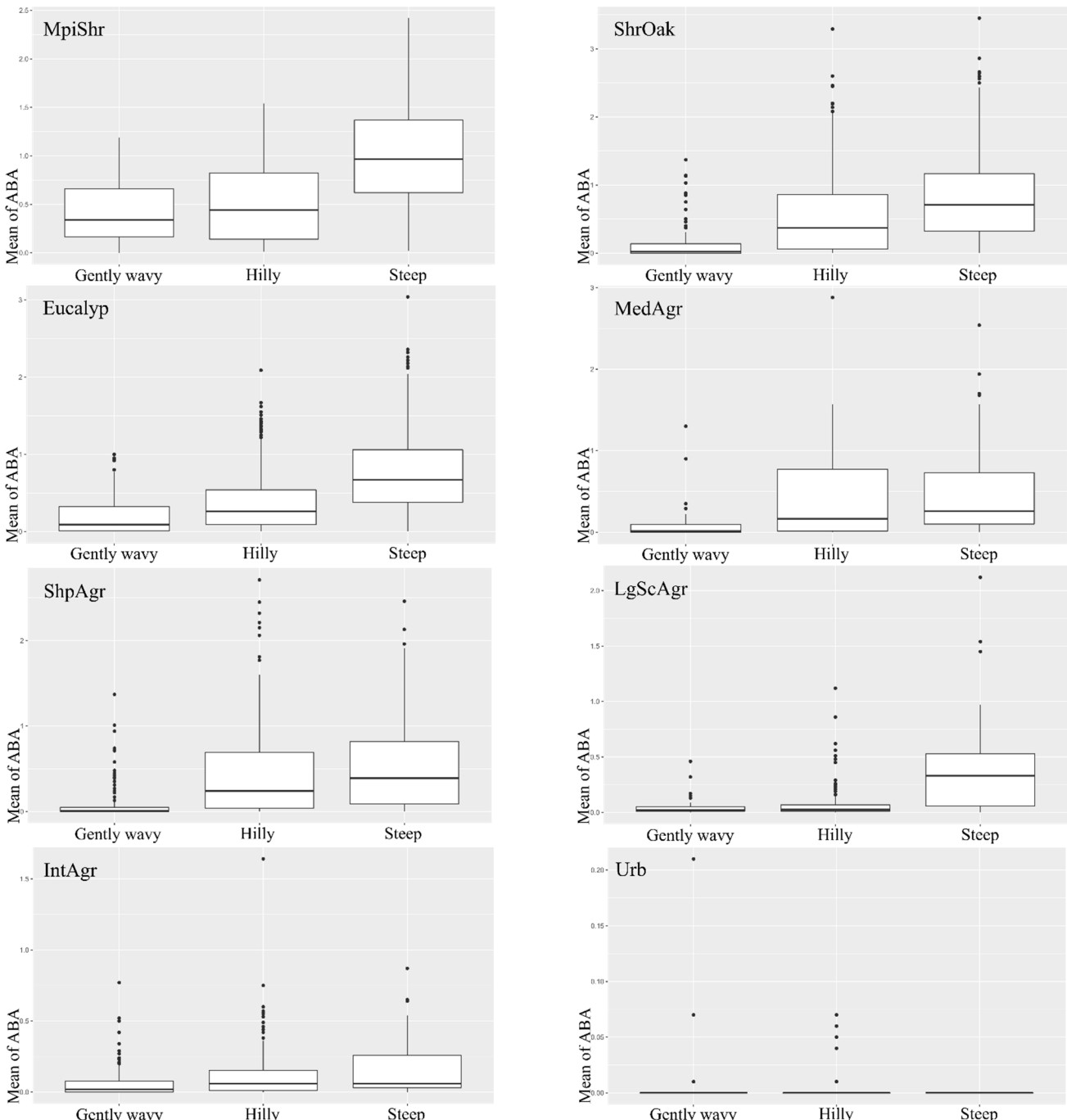

**Figure A7.** Box plots showing the effect of land system types (MpiShr: maritime pine forests and shrubland grazed by goats; ShrOak: shrubland and other oaks and hardwood forests; Eucalyp: eucalyptus forests; MedAgr: Mediterranean agriculture; ShpAgr: grazing sheep; LgScAgr: large-scale agriculture; IntAgr: intensive agriculture; Urb: urban areas) on accumulated burned area (1990–2017) (ABA) across the three land morphology types (Gently wavy; Hilly; Steep). Within each box, the middle horizontal lines indicate median values; the upper and lower bounds of the boxes indicate the 25th to the 75th percentile of each type's distribution of values; the vertical extending lines denote adjacent values; and dots refer to observations outside the range of adjacent values.

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
