# Peer review of "Exploring Land System Options to Enhance Fire Resilience under Different Land Morphologies"

_fire, doi:10.3390/fire6100382_

Round 1

Reviewer 1 Report

I congratulate the authors for putting this interesting and informative timely study together. The article is well written, structured and clear. The figures are easy to interpret as well. I only have very minor comments for clarity.

Ln 94: Consider replacing ‘analyze’ with ‘identify’ for clarity.

Ln 468-469: Perhaps this needs rephrasing. The abandonment of fire-stick farming (not the typical agricultural practice) by Aboriginal people in Australia has also led to shrub encroachment (ladder fuel) in woodlands/forests resulting in large landscape fire over the last century (https://doi.org/10.1002/fee.2395).

Ln 507-510: Consider replacing ‘studies’ with ‘a study’ or add one more source to ‘[6]’.

Author Response

Manuscript ID: fire-2644724

Response to Reviewer 1 Comments

Summary: We would like to extend our sincere gratitude for the time and effort you dedicated to reviewing our manuscript. Your thoughtful feedback and constructive comments have been invaluable in improving the quality and rigor of our research.

Reviewer 1: I congratulate the authors for putting this interesting and informative timely study together. The article is well written, structured and clear. The figures are easy to interpret as well. I only have very minor comments for clarity.

Reviewer 1: Ln 94: Consider replacing ‘analyze’ with ‘identify’ for clarity.

Authors response: We appreciate your attention to detail and your efforts to enhance the clarity of our writing. As suggested, we replaced ‘analyze’ with ‘identify’ (L. 94).

Reviewer 1: Ln 468-469: Perhaps this needs rephrasing. The abandonment of fire-stick farming (not the typical agricultural practice) by Aboriginal people in Australia has also led to shrub encroachment (ladder fuel) in woodlands/forests resulting in large landscape fire over the last century (https://doi.org/10.1002/fee.2395).

Authors response: Thank you for your comment regarding the abandonment of fire-stick farming in Australia and its impact on shrub encroachment and landscape fires. While we acknowledge that this is indeed a specific situation in Australia, in our study we focus on the broader context of widespread agricultural abandonment in mountainous regions in southern Europe, followed by shrub encroachment and afforestation with monocultural forests of maritime pine and eucalyptus. While the consequences of agricultural abandonment may vary between regions, our research examines the general trends and challenges associated with this phenomenon in the specific context of southern European mountain landscapes. To clarify, we have rephrased the sentence as follows: “The mountain landscapes of southern European countries such as Greece or Portugal have been subject to recent phenomena of agricultural abandonment and subsequent shrub encroachment and afforestation with monocultural forests of maritime pine and eucalyptus” (L. 470).

Reviewer 1: Ln 507-510: Consider replacing ‘studies’ with ‘a study’ or add one more source to ‘[6]’.

Authors response: We highly appreciate your attention to detail and your efforts to enhance the clarity of our writing. As suggested, we replaced ‘studies’ with ‘a study’ (L. 509).

Reviewer 2 Report

1. I do not think "Portugal" should be included in the keywords.

2. I suggest the humidity should be listed in the basic information of research area.

3. Fig.8 is almost unreadable.

Author Response

Manuscript ID: fire-2644724

Response to Reviewer 2 Comments

Summary: We would like to extend our sincere gratitude for the time and effort you dedicated to reviewing our manuscript. Your thoughtful feedback and constructive comments have been invaluable in improving the quality and rigor of our research.

Reviewer 2: I do not think "Portugal" should be included in the keywords.

Authors response: We appreciative the reviewer for pointing this out. As suggested, we have deleted "Portugal" from the keywords.

Reviewer 2: I suggest the humidity should be listed in the basic information of research area.

Authors response: Thank you for your valuable feedback and suggestions regarding the inclusion of humidity information in the basic research area details. As suggested, we have already addressed this point in the manuscript, precisely on line 115.

Reviewer 2: Fig.8 is almost unreadable.

Authors response: Thank you for your feedback regarding the three-way plot figure. We highly appreciate your input. However, we would like to seek further clarification regarding your comment that it is “almost unreadable”. The three-way plots are generally complex due to the nature of the data and analysis. We understand your concern and would like to address it effectively. To do so, could you please provide more specific details or point out which aspects of the figure you find challenging to read or understand? This would greatly assist us in making necessary improvements to enhance the figure's clarity and comprehensibility.

Reviewer 3 Report

This research investigates whether there is a possibility of transforming the landscape by single modifying LS from a more to a less fire prone. To better understand landscape-fire relationships, the individual and interactive effects of LS and LM on burned area were also analyzed. With the identification of homogeneous areas of fire-proneness, this study aims to establish priority areas for landscape transformation actions and the application of common strategies of landscape planning and management.

The following remarks should be taken into consideration preparing updated version of the paper:

1. In introduction, classified by slope  in “A LM typology, based on three main landforms: valley bottoms; hillslopes, classified by slope; and hilltops” should be deleted.

2.In line 119 , What exactly "the most critical period"means is not clear.

3.The two methods of PCA and HCA are used in the 2.5 statistical analysis, but the introduction of them and how to apply them are missing.

4.In 3.2, it is mentioned that these seven components retained 66% of the variability of the original data, how is the data obtained?

5.In 3.3, why the low p-value can indicates that a statistically significant relationship exists between LM and LS types? And how the bounds of low p-values are determined to be 0.001?

Minor modification.

Author Response

Manuscript ID: fire-2644724

Response to Reviewer 3 Comments

Summary: We would like to extend our sincere gratitude for the time and effort you dedicated to reviewing our manuscript. Your thoughtful feedback and constructive comments have been invaluable in improving the quality and rigor of our research. As recommended, we have performed minor editing of the English language.

Reviewer 3: This research investigates whether there is a possibility of transforming the landscape by single modifying LS from a more to a less fire prone. To better understand landscape-fire relationships, the individual and interactive effects of LS and LM on burned area were also analyzed. With the identification of homogeneous areas of fire-proneness, this study aims to establish priority areas for landscape transformation actions and the application of common strategies of landscape planning and management. The following remarks should be taken into consideration preparing updated version of the paper:

Reviewer 3: In introduction, “classified by slope” in “A LM typology, based on three main landforms: valley bottoms; hillslopes, classified by slope; and hilltops” should be deleted.

Authors response: Thank you for your valuable feedback. As suggested, we have deleted “classified by slope”.

Reviewer 3: In line 119, What exactly "the most critical period" means is not clear.

Authors response: Thank you for pointing this out. What we intended to emphasize was that the summer period corresponds to the wildfire season. To clarify, we modified the sentence to: “Despite this climatic variability between north and south, Portugal presents the typical characteristics of the Mediterranean climate, with the highest temperatures, lowest relative humidity levels, and strong winds concentrated in the summer period, namely in July, August and September, creating the optimal conditions for fire occurrence.”

Reviewer 3: The two methods of PCA and HCA are used in the 2.5 statistical analysis, but the introduction of them and how to apply them are missing.

Authors response: Thank you for your thoughtful feedback on our manuscript. We appreciate your attention to detail and your suggestions for improvement. As suggested, we have introduced PCA and HCA in Lines 219-237 of the manuscript. In this section, we provide a concise but comprehensive overview of these techniques. Specifically, we explain that PCA is employed for dimensionality reduction, while HCA using Ward's method is utilized for clustering analysis in our study. We believe that these additions now provide a clearer understanding of the methods used in our research. If you feel that further clarification or expansion is needed in the introduction of these techniques, please do not hesitate to let us know. We are committed to ensuring that our manuscript is as informative and well-structured as possible.

Reviewer 3: In 3.2, it is mentioned that these seven components retained 66% of the variability of the original data, how is the data obtained?

Authors response: Thank you for your question regarding the methodology used to obtain the seven principal components that retained 66% of the variability in the original data. In our analysis, we applied the eigenvalue greater than 1 criterion to determine which principal components to retain. To clarify the process, we performed a principal component analysis (PCA) on the dataset. In PCA, the eigenvalues represent the amount of variance explained by each principal component. We retained the principal components with eigenvalues greater than 1 because they individually explained more variance than would be expected by chance. The eigenvalues are obtained as part of the PCA output, and those exceeding 1 were included in our final analysis. This method allowed us to capture a significant portion of the data's variability while reducing dimensionality. If you require further details or have additional questions regarding our methodology, please feel free to ask. Your feedback is greatly appreciated.

Reviewer 3: In 3.3, why the low p-value can indicates that a statistically significant relationship exists between LM and LS types? And how the bounds of low p-values are determined to be 0.001?

Authors response: We appreciate your insightful questions regarding the statistical analysis in our manuscript. We would like to address your concerns regarding the interpretation of p-values and the choice of a significance level of 0.001. Low p-values, such as those obtained in our analysis, typically indicate that the observed relationship between LM types and LS types is unlikely to have occurred by chance, assuming the null hypothesis. In our case, a low p-value (p value < 0.05) suggests a statistically significant relationship between these classifications.

We would like to clarify that the methodology used in the analysis is described in lines 238-252 of the manuscript, where we provide a comprehensive account of the chi-square test, Kruskal-Wallis test, Dunn's test, and two-way analysis of variance (ANOVA). We hope this clarifies the methodology and rationale behind our statistical analysis. If you require further information or have additional questions, please do not hesitate to ask. We are committed to ensuring the rigor and clarity of our analysis.
